# Causal Estimation of Exposure Shifts with Neural Networks: Evaluating the Health Benefits of Stricter Air Quality Standards in the US

## Abstract

In policy research, one of the most critical analytic tasks is to estimate the causal effect of a policy-relevant shift to the distribution of a continuous exposure/treatment on an outcome of interest. We call this problem *shift-response function* (SRF) estimation. Existing neural network methods involving robust causal-effect estimators lack theoretical guarantees and practical implementations for SRF estimation. Motivated by a key policy-relevant question in public health, we develop a neural network method and its theoretical underpinnings to estimate SRFs with robustness and efficiency guarantees. We then apply our method to data consisting of 68 million individuals and 27 million deaths across the U.S. to estimate the causal effect from revising the US National Ambient Air Quality Standards (NAAQS) for $PM_{2.5}$ from 12 $\mu g/m^3$ to 9 $\mu g/m^3$. This change has been recently proposed by the US Environmental Protection Agency (EPA). Our goal is to estimate, for the first time, the reduction in deaths that would result from this anticipated revision using causal methods for SRFs. Our proposed method, called **T**argeted **R**egularization for **E**xposure **S**hifts with Neural **Net**works (TRESNET), contributes to the neural network literature for causal inference in two ways: first, it proposes a targeted regularization loss with theoretical properties that ensure double robustness and achieves asymptotic efficiency specific for SRF estimation; second, it enables loss functions from the exponential family of distributions to accommodate non-continuous outcome distributions (such as hospitalization or mortality counts). We complement our application with benchmark experiments that demonstrate TRESNET's broad applicability and competitiveness.

## 1 Introduction

The field of causal inference has seen immense progress in the past couple of decades with the development of *targeted* doubly-robust methods yielding desirable theoretical efficiency guarantees on estimates of various causal effects (Van der Laan et al., 2011; Kennedy, 2016). These advancements have been recently incorporated into the neural network (NN) literature for causal inference via *targeted regularization* (TR) (Shi et al., 2019; Nie et al., 2021). TR methods produce favorable properties for causal estimation by incorporating a regularization term into a supervised neural network model. However, it remains an open task to develop a NN method that specifically targets the causal effect of a shift in the distribution for a continuous exposure/treatment variable (Muñoz & Van Der Laan, 2012). We call this problem *shift-response function* (SRF) estimation. Many scientific questions can be formulated as an SRF estimation task (Muñoz & Van Der Laan, 2012). Some notable examples include estimating the health effects of shifts in the distribution of environmental, socioeconomic, and behavioral variables (e.g., air pollution, income, exercise habits) (Muñoz & Van Der Laan, 2012; Díaz & Hejazi, 2020; Smith et al., 2023).

Our objective is to develop a neural network technique that addresses a timely and highly prominent regulatory question. More specifically, the EPA is currently considering whether or not to revise the National Ambient Air Quality Standards (NAAQS), potentially lowering the permitted annual-average $PM_{2.5}$ concentration from 12 to 11, 10 or 9 $\mu g/m^3$. We anticipate that the revision of the

NAAQS will ultimately result in a shift to the distribution of $PM_{2.5}$ concentrations. Our goal is to estimate, for the first time, the reduction in deaths that would result from this anticipated shift using causal methods for SRFs.

**Contributions** We contribute to the public debate informing the US Environmental Protection Agency (EPA) on the effects of modifying air quality standards. A preview of the results (fully developed in Section 6) is presented in Figure 1. The figure presents the estimated reduction in deaths (%) resulting from various shifts to the distribution of $PM_{2.5}$ across every ZIP-code in the contiguous US between 2000 and 2016. These shifts limit the maximum concentration of $PM_{2.5}$ to the cutoff value for every ZIP-code that exceeds the cutoff. We vary the cutoff in this SRF between 6 $\mu g/m^3$ and 16 $\mu g/m^3$ (x-axis). The y-axis represents the % reduction in deaths that corresponds with each cutoff threshold. Notably, a NAAQS threshold of 9 $\mu g/m^3$, would have had the effect of decreasing elder mortality by 4%. These findings present a data-driven perspective on the potential health benefits of the EPA's proposal.

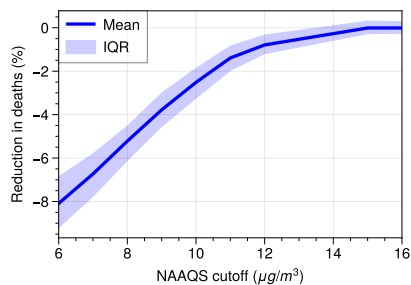

Figure 1: Estimated mortality reduction under a *cutoff exposure shift* lowering the annual $PM_{2.5}$ in all regions below a given threshold. Uncertainty bands represent the interquartile range from an ensemble of networks. *Data source*: US Medicare claims from 2000–2016.

In our implementation of this analysis, we present a novel method, called **T**argeted **R**egularization for **E**xposure **S**hifts with Neural **Net**works (TRESNET), which introduces two necessary and generalizable methodological innovations to the TR literature. First, we use a TR loss that specifically targets SRFs, ensuring that our estimates retain the properties we have come to expect from TR methods such as asymptotic efficiency and double robustness (Kennedy, 2016). Given standard regularity conditions, these guarantees imply that the SRF is consistently estimated when either the outcome model or the density-ratio model for the exposure shift is correctly specified, and achieves the best possible efficiency rate when both models are correctly specified. Second, TRESNET accommodates non-continuous outcomes belonging to the exponential family of distributions (such as mortality counts) that frequently arise in real-world scenarios, including our motivating application. In addition to its suitability for our application, we assess the performance of TRESNET in a simulation study tailored for SRF estimation, demonstrating improvements over neural network methods not specifically designed for SRFs.

**Related work** Recent papers have most often estimated causal effects relating air pollution to elder mortality using exposure-response functions (ERFs); see for example Wu et al. (2020); Bahadori et al. (2022); Josey et al. (2023). However, none of the methods implemented in these studies target an SRF estimand. We elaborate on the distinction between ERFs and SRFs in Section 2, emphasizing the latter's importance for our motivating application and informing policy.

Neural network-specific methods for causal inference are divided between works that aim to estimate individualized effects (e.g., Bica et al. (2020); Yoon et al. (2018)) and those targeting marginal effects from a population with efficiency guarantees derived from a specific causal estimand. Several methods in the latter category—including this work—are based on deriving estimating equations using a functional quantity unique to the target estimand known as the *efficient influence function* (EIF). EIFs have been widely studied for deriving asymptotically efficient estimators with doubly robust properties (Kennedy, 2016; Bang & Robins, 2005; Robins, 2000; Bickel et al., 1993). EIFs are also referred to as Neyman orthogonal scores in the double machine learning literature (Kennedy, 2022). *Targeted regularization* (TR) (Shi et al., 2019) links EIF estimation methods to neural network architectures and optimization. Recent uses of TR include the DRAGONNET (Shi et al., 2019), which introduced TR for targeting the average treatment effect of a binary treatment, and the VCNET (Nie et al., 2021), for targeting the exposure-response function (ERF) of a continuous exposure.

Outside of the the neural network literature, SRFs have been studied in the *stochastic intervention* and *modified treatment policy* frameworks for causal inference (Muñoz & Van Der Laan, 2012; Díaz et al., 2021). Insights gained from the advancements in this space have guided the formulation of our applied research question and our implementation of a neural network architecture.

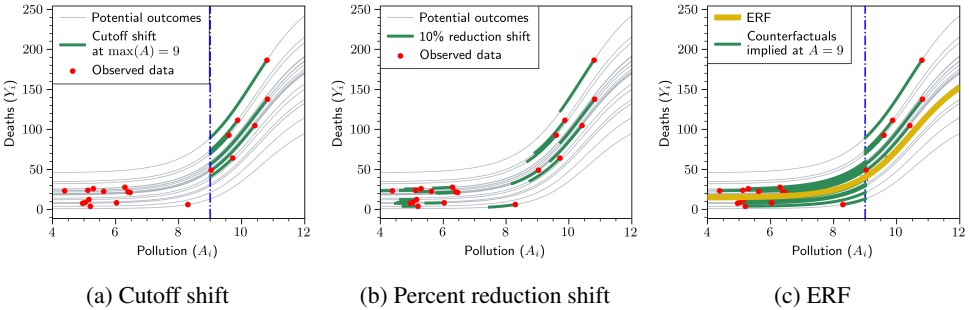

| (a) Cutoff shift | (b) Percent reduction shift | (c) ERF |

Figure 2: Two examples of exposure shifts with their implied counterfactuals and, for comparison, the implied counterfactuals of an exposure-response function at a given exposure value.

## 2 PROBLEM STATEMENT: THE CAUSAL EFFECT OF AN EXPOSURE SHIFT

The notation we use follows standard language and concepts of the potential outcomes framework (Imbens & Rubin, 2015). Let $(A, Y, \boldsymbol{X})$ denote a unit from the target population $\mathbb{P}$, where $A \in \mathcal{A}$ is a continuous exposure/treatment variable, $Y \in \mathcal{Y}$ is the outcome of interest, and $\boldsymbol{X} \in \mathcal{X}$ are covariates. Suppose we obtain a sample of $n$ iid observations $\{\boldsymbol{X}_i, A_i, Y_i\}_{i=1}^n$. The *potential outcomes* notation $Y^a$ represents the outcome corresponding to the exposure/treatment level $a$. The potential outcomes for unobserved exposure values are called *counterfactuals* and *factuals* otherwise. The *consistency* condition requires that the factual outcomes satisfy $Y^A = Y$ for the observed treatment $A$. The *no interference* condition requires that the exposure of one unit does not cause the potential outcomes of another unit [1].

The conditional expectation of the potential outcomes is defined as $\mu(\boldsymbol{x}, a) = \mathbb{E}[Y^a | \boldsymbol{X} = \boldsymbol{x}]$. All expectations are with respect to $\mathbb{P}$ unless stated otherwise. For conciseness, we also use a generic notation $p$ to denote the density function of the random variables composing the data distribution. For example, $\boldsymbol{X} \sim p(\boldsymbol{X})$, $A | \boldsymbol{X} \sim p(A | \boldsymbol{X})$, and so on. The latter quantity $p(A | \boldsymbol{X})$ is known as the *generalized propensity score*.

**Estimand of interest: effects of exposure shifts**   An *exposure shift* represents a counterfactual scenario in which an intervention modifies a unit's exposure, usually in reference to its observed value. It considers the case when the exposure data consists of pairs $(A, \tilde{A})$ in which $A$ is the exposure observed in the data and $\tilde{A}$ indicates the exposure that would be observed after the shift. As simple examples, illustrated in Figure 2, one can define a cutoff shift $\tilde{A} = \min\{A, c\}$ where the exposure is truncated to a maximum value of $c$; another example is $\tilde{A} = cA$, in which the exposure is multiplied by a factor $c$. For instance, $c = 0.9$ would represent a 10% reduction to all units with respect to its observed value. We will denote the shifted generalized propensity score as $\tilde{p}(\tilde{A} | \boldsymbol{X})$.

The shift-response function (SRF) estimand, denoted $\psi = \psi(\mathbb{P})$, is the expected potential outcome induced by the exposure shift:

$$\psi = \mathbb{E}_{\boldsymbol{X} \sim p(\boldsymbol{X})} \left[ \mathbb{E}_{\tilde{A} \sim \tilde{p}(\tilde{A} | \boldsymbol{X})} \left[ \mu(\boldsymbol{X}, \tilde{A}) \mid \boldsymbol{X} \right] \right] = \mathbb{E}[Y^{\tilde{A}}]. \tag{1}$$

This estimand cannot be expressed in terms of traditional causal effects such as the average treatment effect (ATE) or an exposure-response function (ERF) (Muñoz & Van Der Laan, 2012).

There is no restriction on how $\tilde{A}$ is defined as long as pairs $(A, \tilde{A})$ are available. The shift can be a stochastic or deterministic function $\tilde{A} = f(A, \boldsymbol{X})$ (Díaz et al., 2021). Importantly, we do not require $\tilde{p}$ to be known, nor do we need a deterministic formula expressing $\tilde{A}$ in terms of $A$.

**Causal identification**   The target estimand $\psi$ can be expressed as a functional of the observable data distribution under standard assumptions, which are:

---

[1]*Consistency* and *no interference* encompass the *the stable unit treatment value assignment* (SUTVA) assumption, which is a standard structural assumption used to formalize causal estimation problems in the potential outcomes framework (Imbens & Rubin, 2015).

**Assumption 2.1** (Unconfoundedness). $A \perp\!\!\!\perp Y^a \mid \boldsymbol{X}$ for all $a \in \mathcal{A}$.

**Assumption 2.2** (Positivity). Let $w(\boldsymbol{x}, a) = \tilde{p}(a|\boldsymbol{x})/p(a|\boldsymbol{x})$. Then, there exist a constant $M > 0$ such that $w(\boldsymbol{x}, a) < M$ for all $(a, \boldsymbol{x})$ such that $p(a|\boldsymbol{x}) > 0$.

The first assumption ensures that $\mu(\boldsymbol{x}, a) = \mathbb{E}[Y|\boldsymbol{X} = \boldsymbol{x}, A = a]$. The right-hand side of this equality can be directly estimated via regression whereas the left-hand side cannot. The second assumption implies that the density ratio $w(\boldsymbol{x}, a)$ is well-defined and behaved. Notice that $\psi = \mathbb{E}[\mu(\boldsymbol{X}, A)w(\boldsymbol{X}, A)]$ by the importance sampling formula. Therefore, estimators of $\mu$ and $w$ will suffice to estimate $\psi$. Intuitively, Assumption 2.2 prohibits extreme cases when the shifted exposures take value outside the practical domain of the observed exposure, in which case counterfactual estimation is impossible (Muñoz & Van Der Laan, 2012).

**Mutiple shifts** We can estimate the effect of multiple exposure shifts simultaneously. Let $\tilde{p} \in \tilde{\mathcal{P}}$ denote the set of finite exposure shifts of interest. We can index the density ratio and estimand by $\tilde{p}$, and denote $\boldsymbol{w} = (w_{\tilde{p}})_{\tilde{p} \in \tilde{\mathcal{P}}}$, $\boldsymbol{\psi} = (\psi_{\tilde{p}})_{\tilde{p} \in \tilde{\mathcal{P}}}$ and $\tilde{\boldsymbol{A}} = (\tilde{A}^{\tilde{p}})_{\tilde{p} \in \tilde{P}}$.

**Comparison with ERFs** Exposure-response functions are common estimands in the causal inference literature for continuous treatments. Mathematically, the ERF $\xi$ can be written as the mapping $\xi(a) = \mathbb{E}[\mu(\boldsymbol{X}, A)|A = a]$. One can consider ERFs as a limiting case of SRFs when $\tilde{p}$ is a point mass distribution centered at a fixed treatment value assigned *equally to all units*. A visual example is shown in Figure 2c. The fundamental reason why SRFs are more suitable for our motivating application is that ERFs do not allow us to consider scenarios like the cutoff shift, illustrated in Figure 1. In this setting the $PM_{2.5}$ levels are reduced only for locations that did not comply with the proposed air quality standard. An ERF describes the average outcome when *all* units are given the *same* $PM_{2.5}$ value. Thus, theoretical guarantees for ERF estimation do not apply to SRFs. Moreover, our experiments suggest that the TR estimators designed for ERFs hampers estimation of SRFs.

# 3 TRESNET: TARGETED REGULARIZATION FOR ESTIMATING THE CAUSAL EFFECTS OF EXPOSURE SHIFTS WITH NEURAL NETWORKS

As we have suggested earlier, an estimator of $\boldsymbol{\psi}$ can be derived from estimators of the outcome and density ratio functions. Using TR, we will obtain an estimator $\hat{\boldsymbol{\psi}}^{\text{tr}}$ with the guarantees that $\|\hat{\boldsymbol{\psi}}^{\text{tr}} - \boldsymbol{\psi}\|_2$ converges in probability at an "efficient" rate according to the prevailing semiparametric efficiency theory surrounding robust causal effect estimation (Kennedy, 2022).

**The efficient influence function (EIF)** The EIF of $\boldsymbol{\psi}$, denoted $\varphi(\boldsymbol{O}; \boldsymbol{\psi}, \mu, \boldsymbol{w})$, is a fundamental function in the theory of semiparametric models (Kennedy, 2022). More concisely, the EIF is the gradient of $\boldsymbol{\psi}$ with respect to small perturbations in the data distribution. Results from semiparametric theory show that the best possible variance among the family of regular, asymptotically linear estimators of $\boldsymbol{\psi}$ is bounded below by $\mathbb{P}[\varphi(\boldsymbol{\psi}, \mu, \boldsymbol{w})\varphi(\boldsymbol{\psi}, \mu, \boldsymbol{w})^{\top}]$. Moreover, the asymptotic variance of any statistically consistent estimator $(\hat{\boldsymbol{\psi}}, \hat{\mu}, \hat{\boldsymbol{w}})$ satisfying the empirical estimating equation (EEE) $\mathbb{P}_n(\varphi(\hat{\boldsymbol{\psi}}, \hat{\mu}, \hat{\boldsymbol{w}})) = 0$ eventually achieves this lower bound. For the SRF, the EIF is given by

$$\varphi(\boldsymbol{O}; \boldsymbol{\psi}, \mu, \boldsymbol{w}) = \boldsymbol{w}(\boldsymbol{X}, A)(Y - \mu(\boldsymbol{X}, A)) + \mu(\boldsymbol{X}, \tilde{\boldsymbol{A}}) - \boldsymbol{\psi}. \tag{2}$$

We provide a proof and additional background about the derivation of the EIF in the appendix. The reader can refer to Tsiatis (2006) and Kennedy (2022) for a more comprehensive introduction to the EIF and semiparametric efficiency theory. Observe that if $(\hat{\boldsymbol{\psi}}, \hat{\mu}, \hat{\boldsymbol{w}})$ satisfies the EEE, then $\hat{\boldsymbol{\psi}}$ can be decomposed in terms of a debiasing component of the residual error and a plugin estimator for the marginalized average of the mean response:

$$\hat{\boldsymbol{\psi}} = \underbrace{\frac{1}{n}\sum_{i=1}^{n} \hat{\boldsymbol{w}}(\boldsymbol{X}_i, A_i)(Y_i - \hat{\mu}(\boldsymbol{X}_i, A_i))}_{\text{debiasing term}} + \underbrace{\frac{1}{n}\sum_{i=1}^{n} \hat{\mu}(\boldsymbol{X}_i, \tilde{\boldsymbol{A}}_i)}_{\text{plugin estimator}} \tag{3}$$

**TR for SRFs** An immediate approach to obtain a doubly-robust estimator satisfying the EEE would be to use the right-hand side of Equation (3) as specifying an estimator from a finite sample and nuisance function estimators $\hat{\mu}$ and $\hat{\boldsymbol{w}}$. Such an estimator, denoted $\hat{\boldsymbol{\psi}}_{\text{aipw}}$, is sometimes called the *augmented inverse-probability weighting* (AIPW) estimator for exposure shifts (also referred to

as modified treatment policies and stochastic interventions) (Muñoz & Van Der Laan, 2012; Díaz et al., 2021). This estimator must be distinguished from the standard AIPW estimator for traditional average causal effects (ATE and ERF) (Robins, 2000; Robins et al., 2000).

TR is an alternative approach based on the observation that the debiasing term in Equation (3) has been empirically observed to affect performance in finite samples due to its sensitivity to $\hat{w}$. Instead, TR learns a perturbed outcomes model using a special regularization loss, ensuring that the resulting plugin estimator (the second component of Equation (3)) satisfies the EEE without requiring the debiasing term. We introduce the perturbation model and regularization loss in the next section.

**Generalized TR for outcomes in the exponential family**   We present a general formulation applicable to the SRF estimand for any outcome supported by a generalized domain. First, we say that the outcome follows a conditional distribution from the exponential family if $p(Y|\boldsymbol{X}, A) \propto \exp(Y\eta(\boldsymbol{X}, A) - \Lambda(\eta(\boldsymbol{X}, A))$ for some function $\eta : \mathcal{X} \times \mathcal{A} \to \mathbb{R}$. The family's canonical link function $g$ is defined by the identity $g(\mathbb{E}[Y \mid \boldsymbol{X}, A]) = \eta(\boldsymbol{X}, A)$. For all distributions in the exponential family, $g$ is invertible (McCullagh, 2019). Exponential families allow us to consider the usual mean-squared error and logistic regression as special cases. They also enable modeling of death counts as in our application analyzing the health effects of $PM_{2.5}$. In this setting, we set $\Lambda(\eta) = e^\eta$ and $g(\mu) = \log(\mu)$; a Poisson regression environment.

The following theorem forms the basis for the TR estimator.

**Theorem 1.** *Let $\epsilon$ denote a perturbation parameter and define*

$$\mathcal{L}^{tr}(\mu^{NN}, \boldsymbol{w}^{NN}, \epsilon)(\boldsymbol{O}) = \Lambda(g(\mu^{NN}(\boldsymbol{X}, A)) + \epsilon) - (g(\mu^{NN}(\boldsymbol{X}, A)) + \epsilon)Y.$$
$$\mathcal{R}^{tr}(\mu^{NN}, \boldsymbol{w}^{NN}, \epsilon) = \frac{1}{n}\sum_{i=1}^{n} \mathcal{L}^{tr}(\mu^{NN}, \boldsymbol{w}^{NN}, \epsilon)(\boldsymbol{O}_i). \tag{4}$$

*Then $(\frac{\partial \mathcal{R}^{tr}}{\partial \epsilon})(\mu^{NN}, \boldsymbol{w}^{NN}, \epsilon) = 0$ iff $\frac{1}{n}\sum_{i=1}^{n} \boldsymbol{w}^{NN}(\boldsymbol{X}_i, A_i)(Y_i - g^{-1}(g(\mu^{NN}(\boldsymbol{X}_i, A_i)) + \epsilon))) = 0$.*

The condition $\partial \mathcal{R}^{tr}/\partial \epsilon = 0$ in the theorem holds upon minimization of $\mathcal{R}^{tr}$. Consequently, the TR estimator $\hat{\psi}^{tr}$ is defined as the solution of an optimization problem

$$(\hat{\mu}, \hat{\boldsymbol{w}}, \hat{\epsilon}) = \underset{\mu^{NN}, \boldsymbol{w}^{NN}, \epsilon}{\arg\min} \mathcal{R}_\mu(\mu^{NN}) + \alpha\mathcal{R}_{\boldsymbol{w}}(\boldsymbol{w}^{NN}) + \beta_n\mathcal{R}^{tr}(\mu^{NN}, \boldsymbol{w}^{NN}, \epsilon)$$
$$\hat{\psi}^{tr} := \frac{1}{n}\sum_{i=1}^{n} g^{-1}(g(\hat{\mu}(\boldsymbol{X}, A)) + \hat{\epsilon})) \tag{5}$$

where $\mathcal{R}_\mu$ and $\mathcal{R}_{\boldsymbol{w}}$ are the empirical risk functions of $\mu$ and $\boldsymbol{w}$, $\alpha > 0$ is a hyperparameter, and $\beta_n$ is a regularization weight satisfying $\beta_n \to 0$. The latter condition is needed to ensure statistical consistency, as first discussed by Nie et al. (2021) for ERF estimation. Section 4 provides additional details about the architecture and risk function specification. The full loss in Equation (5) preserves the fact that $\frac{\partial \mathcal{R}^{tr}}{\partial \epsilon} = 0$ since $\epsilon$ only appears in the regularization term.

**Double robustness and efficiency**   Before introducing the main result of the TR estimator for the SRF, we require the following additional notation. For any $f : \mathcal{U} \to \mathbb{R}$, $\|f\|_\infty = \sup_{u \in \mathcal{U}} |f(u)|$ and $\|\mathcal{F}\|_\infty = \sup_{f \in \mathcal{F}} \|f\|_\infty$. We define the sample Rademacher complexity as $\text{Rad}_n(\mathcal{F}) = \sup_{f \in \mathcal{F}} |\frac{1}{n}\sum_{i=1}^{n} \sigma_i f(U_i)|$ where $\sigma_i$ are iid Rademacher random variables satisfying $p(\sigma_i = 1) = p(\sigma_i = -1) = 1/2$. It is easiest to think of the Rademacher complexity as a natural measure for the degrees of freedom of an estimating class of functions. We use $O_p$ and $o_p$ to denote stochastic boundedness and convergence in probability, respectively.

**Theorem 2.** *Let $\mathcal{M}$ and $\mathcal{W}$ be classes of functions such that $\hat{\mu}, \mu \in \mathcal{M}$ and $\hat{\boldsymbol{w}}, \boldsymbol{w} \in \mathcal{W}$. Suppose assumptions 2.1 and 2.2 hold, and that the following regularity conditions hold: (i) $\|\mathcal{M}\|_\infty < \infty$, $\|\mathcal{W}\|_\infty < \infty$, $\|1/\mathcal{W}\|_\infty < \infty$; (ii) either $\hat{\mu} = \mu$, $\hat{\boldsymbol{w}} = \boldsymbol{w}$, or $\text{Rad}_n(\mathcal{M}) = O(n^{-1/2})$ and $\text{Rad}_n(\mathcal{W}) = O(n^{-1/2})$; (iii) the loss function in Equation (4) is Lipschitz; (iv) $\Lambda$ and $g$ are twice continuously differentiable. Then, the following statements are true:*

1. *The outcome and density ratio estimators of TR are consistent. That is, $\hat{\mu} \xrightarrow{p} \mu$ and $\hat{\boldsymbol{w}} \xrightarrow{p} \boldsymbol{w}$.*
2. *The estimator $\hat{\psi}^{tr}$ satisfies $\|\hat{\psi}^{tr} - \psi\|_\infty = O_p(n^{-1/2} + r_1(n)r_2(n))$ whenever $\|\hat{\mu} - \mu\|_\infty = O_p(r_1(n))$ and $\|\hat{\boldsymbol{w}} - \boldsymbol{w}\|_\infty = O_p(r_2(n))$.*

Theorem 2 shows that the TR regularized learner of the SRF achieves "optimal" root-$n$ convergence when $r_1(n) = r_2(n) = n^{-1/4}$ or when either $r_1$ or $r_2$ vanishes. Using standard arguments involving concentration inequalities, the Lipschitz assumption on the loss function can be relaxed by assuming that the loss function has a vanishing Rademacher complexity (Wainwright, 2019).

## 4 ARCHITECTURE FOR ESTIMATING $\mu$ AND $w$ WITH NEURAL NETWORKS

We describe a simple yet effective architecture for estimating $\mu$ and $w$. To keep the notation simple, we will write $f_\theta$ to denote generic output from a neural network indexed by weights $\theta$. The architecture has three components, illustrated in Figure 3. The first component maps the confounders $X$ to a latent representation $Z = f_{\theta_Z}(X) \in \mathbb{R}^d$. This component will typically be a multi-layer perception (MLP). The second and third components are the outcome and density-ratio heads, which are functions of $Z$ and the treatment, respectively. We describe all three components in detail below.

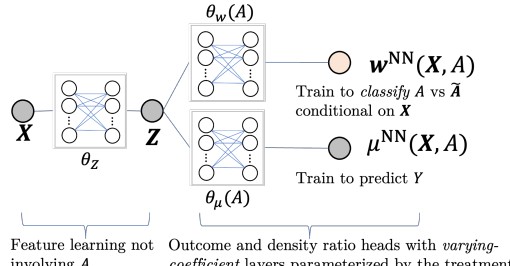

Figure 3: TRESNET architecture using a head for the density ratio model and a head for the outcome model.

Neural network architectures for nuisance function estimation have been widely investigated in causal inference; see Farrell et al. (2021) for a review. We use the architectures proposed in the TR literature as a building block, particularly for continuous treatments (Nie et al., 2021). Nonetheless, It must be remarked that previous work in TR has not yet investigated architectures required for SRF estimation. In particular, we need a new architecture to estimate the density ratio $w$. Previous works have only focused on architectures for estimating propensity scores as required by traditional causal effect estimation (ATEs and ERFs).

**Outcome model** Recall that we assume the outcome $Y$ follows a conditional distribution from the exponential family. That is, $p(Y|X, A) \propto \exp(Y\eta(X, A) - \Lambda(\eta(X, A)))$ with an invertible link function $g$ satisfying $\mu(X, A) = g^{-1}(\eta(X, A))$. We can identify the canonical parameter $\eta$ with the output of the neural network and learn $\hat\mu$ by minimizing the empirical risk

$$\mathcal{R}_\mu(\mu^{\text{NN}}) = \frac{1}{n}\sum_{i=1}^n \left\{ \Lambda(g^{-1}(\mu^{\text{NN}}(Z, A))) - Yg^{-1}(\mu^{\text{NN}}(Z, A)) \right\}. \tag{6}$$

Next, we need to select a functional form for the neural network. An MLP parameterization with the concatenated inputs of $(Z, A)$–the naïve choice–would likely result in the effect of $A$ being lost in intermediate computations. Instead, we adopt the *varying coefficient* approach by setting $\mu^{\text{NN}}(X, A) = g^{-1}(f_{\theta_\mu(A)}(Z))$ (Nie et al., 2021; Chiang et al., 2001). With this choice, the weights of each layer are dynamically computed as a function of $A$ obtained from a linear combination of basis functions spanning the set of admissible functions on $A$. The weights of the linear combination are themselves a learnable linear combination of the hidden outputs from the previous layer. We refer the reader to Nie et al. (2021) for additional background on varying-coefficient layers. Our experiments suggest TR is beneficial for different choices of basis functions.

**Estimation of $w$ via classification** The density ratio head $w^{\text{NN}}$ is trained using an auxiliary classification task. The goal is to estimate the density ratio $w_j$ for each $j = 1, \ldots, |\tilde{P}|$. For this purpose, we use an auxiliary classification task where the positive labels are assigned to the samples from $\tilde{A}_j$ and the negative labels to the samples with $A$ such that

$$\mathcal{R}_w(w^{\text{NN}}) = \sum_{i=1}^n \sum_{\tilde{p} \in \tilde{P}} \frac{1}{2n|\tilde{P}|} \left\{ \text{BCE}(\log w_{\tilde{p}}^{\text{NN}}(X, \tilde{A}_{i\tilde{p}}), \mathbf{1}) + \text{BCE}(\log w_{\tilde{p}}^{\text{NN}}(X_i, A_i), \mathbf{0}) \right\} \tag{7}$$

where BCE stands for the binary cross-entropy classification loss. Equation (7) is a multi-head neural network version of the loss proposed in Díaz et al. (2021) to estimate the effects of modified treatment policies. To capture the role of $A$ more accurately, we propose to parameterize the network using the varying-coefficient structure discussed in the previous section with $\log w_{\tilde{p}}^{\text{NN}}(X, A) = f_{\theta_w^{\tilde{p}}(A)}(Z)$. To our knowledge, we are the first to consider a varying-coefficient architecture for density ratio estimation.

## 5 SIMULATION STUDY

We conducted simulation experiments to validate the design choices of TRESNET.

| | SPLINE-BASED VARYING COEFFICIENTS | | | | PIECEWISE LINEAR VARYING COEFFICIENTS | | | |
| DATASET | $\text{AIPW}_{\text{VC}}$ | $\text{OUTCOME}_{\text{VC}}$ | $\text{TRESNET}_{\text{VC}}$ | VCNET | $\text{AIPW}_{\text{PL}}$ | DRNET | $\text{TRESNET}_{\text{PL}}$ | $\text{DRNET+TR}_{\text{ERF}}$ |
|---|---|---|---|---|---|---|---|---|
| IHDP | 3.15 (0.37) | 2.19 (0.06) | **0.61 (0.03)** | 0.63 (0.03) | 1.18 (0.14) | 2.36 (0.06) | **0.15 (0.02)** | 0.19 (0.02) |
| NEWS | 1.5 (0.19) | 3.65 (0.04) | **0.18 (0.02)** | 0.28 (0.03) | 0.99 (0.12) | 0.99 (0.1) | **0.17 (0.01)** | 0.26 (0.03) |
| SIM-B | 4.1 (0.57) | 0.5 (0.05) | **0.26 (0.03)** | 0.29 (0.04) | 1.46 (0.2) | 1.6 (0.2) | **0.14 (0.02)** | 0.16 (0.02) |
| SIM-N | 5.69 (0.64) | 0.52 (0.05) | **0.32 (0.02)** | **0.32 (0.03)** | 1.81 (0.25) | 0.95 (0.06) | **0.14 (0.01)** | 0.15 (0.01) |
| TCGA-1 | 1.13 (0.08) | 0.63 (0.02) | **0.8 (0.01)** | 0.87 (0.03) | 0.76 (0.05) | 0.62 (0.02) | **0.61 (0.02)** | 0.69 (0.03) |
| TCGA-2 | 0.76 (0.09) | 0.24 (0.02) | **0.18 (0.01)** | 0.24 (0.02) | 0.36 (0.05) | 0.17 (0.01) | **0.12 (0.0)** | 0.16 (0.01) |
| TCGA-3 | 0.83 (0.11) | 0.38 (0.03) | **0.1 (0.01)** | 0.15 (0.02) | 0.59 (0.06) | 0.59 (0.04) | **0.08 (0.01)** | 0.15 (0.02) |

(a) Performance of TRESNET for two baseline architectures, including comparisons with VCNET and DRNET.

| experiment | $\text{OUTCOME}_{\text{VC}}$ W/POISSON LOSS | $\text{OUTCOME}_{\text{VC}}$ W/MSE LOSS | $\text{TRESNET}_{\text{VC}}$ W/POISSON LOSS | $\text{TRESNET}_{\text{VC}}$ W/MSE LOSS |
|---|---|---|---|---|
| IHDP | **18.82 (3.18)** | 3726.92 (357.31) | **2.04 (0.08)** | 10986.43 (211.99) |
| NEWS | **3.41 (0.25)** | 372.24 (52.02) | **0.33 (0.05)** | 1187.94 (100.8) |
| SIM-B | **1222.61 (1269.53)** | 8433.98 (1327.22) | **1113.58 (1270.49)** | 16902.41 (1233.54) |
| SIM-N | **50.72 (4.45)** | 2491.63 (310.36) | **4.6 (0.22)** | 18062.85 (243.2) |
| TCGA-1 | **184.89 (6.67)** | 6682.91 (474.32) | **40.56 (19.2)** | 16307.15 (232.64) |
| TCGA-2 | **48.3 (1.44)** | 5854.08 (483.41) | **398.82 (143.96)** | 16112.02 (186.99) |
| TCGA-3 | **18.27 (2.73)** | 14381.06 (516.25) | **12.92 (2.3)** | 8565.17 (447.13) |

(b) Performance of Poisson-based regularization when the true data is Poisson count data

Table 1: Experiment results. The table shows the $\sqrt{\text{MISE}}$ across 100 random seeds with 95% confidence intervals computed with the asymptotic normal formula.

**Synthetic and semi-synthetic benchmark datasets** The so-called fundamental problem of causal inference is that the counterfactuals are never observed in real data. Thus, we need to rely on widely used semi-synthetic datasets to evaluate the validity of our proposed estimators. First, we consider two datasets introduced by Nie et al. (2021), which are continuous-treatment adaptions to the popular datasets IHDP (Hill, 2011) and NEWS (Newman, 2008). We also use Nie et al. (2021)'s fully simulated data, SIM-N. In addition, we consider the fully simulated dataset described in Bahadori et al. (2022), which features a continuous treatment and has been previously used for calibrating models in air pollution studies. Finally, we consider three variants of the TCGA dataset presented in Bica et al. (2020). The three variants consist of three different dosage specifications as the treatment assignment and the corresponding dose-response as the outcome. The datasets described here have been employed without substantial modifications from the original source studies to facilitate fair comparison. Note that for each of these synthetic and semi-synthetic datasets, we have access to the true counterfactuals, which allows us to compute SRFs exactly. We will consider the estimation task of 20 equally-spaced percent reduction shifts between 0-50% from the current observed exposures. More specifically, $\tilde{A} = (1-c)A$ for values of $c$ in $0-50\%$.

**Evaluation metric and task** Given a semi-synthetic dataset $\mathcal{D}$, consider an algorithm that produces an estimator $\hat{\psi}_{\tilde{p},\mathcal{D}}^{(s)}$ of $\psi_{\tilde{p},\mathcal{D}}^{(s)}$ given an exposure shift $\tilde{p}$ and random seed $s$. To evaluate the quality of the estimator, we use the *mean integrated squared error* $\text{MISE}_{\mathcal{D}} = \frac{1}{n_{\text{seeds}}|\tilde{\mathcal{P}}|} \sum_s \sum_{\tilde{p}} |\hat{\psi}_{\tilde{p},\mathcal{D}}^{(s)} - \psi_{\tilde{p},\mathcal{D}}^{(s)}|^2$. This metric is a natural adaption of an analogous metric commonly used in dose-response curve estimation (Bica et al., 2020).

**Experiment 1: Does targeting SRFs help?** We evaluate two variants of TRESNET against alternative SRF estimators, including VCNET (Nie et al., 2021) and DRNET (Schwab et al., 2020), two prominent methods used in causal TR estimation. The first variant of TRESNET uses varying-coefficient layers based on splines—see the discussion in Section 4 for background. We compare this variant, named $\text{TRESNET}_{\text{VC}}$, with the following baselines: (a) VCNET, which uses a similar architecture and plugin estimator as $\text{TRESNET}_{\text{VC}}$, but with a TR designed for ERFs; (b) $\text{AIPW}_{\text{VC}}$ which is the doubly robust, augmented inverse probability weighting (AIPW) estimator for SRFs (Muñoz & Van Der Laan, 2012) wherein we fit separate outcome and density ratio models that are then substituted into Equation (3); and (c) $\text{OUTCOME}_{\text{VC}}$, which is the same as $\text{TRESNET}_{\text{VC}}$ but without the TR and density ratio heads. The second variant of TRESNET uses varying coefficients based on piecewise linear functions instead of splines. This variant, $\text{TRESNET}_{\text{PL}}$, is compared against analogous baselines: (a) $\text{AIPW}_{\text{PL}}$ for the AIPW estimator; (b) DRNET, which is similar to $\text{OUTCOME}_{\text{VC}}$ with piecewise linear basis functions; and (c) $\text{DRNET} + \text{TR}_{\text{ERF}}$, which uses the same regularization loss for ERFs as VCNET.

Table 1b shows the results of this experiment. For both architectures, the TRESNET variants achieve the best performance. TRESNET$_{\text{VC}}$ is somewhat better than VCNET, which have comparable architectures although TRESNET uses a different TR implementation specific for SRF estimation rather than ERF estimation. Likewise, TRESNET$_{\text{PL}}$ outperforms DRNET + TR$_{\text{ERF}}$. These moderate but consistent performance gains suggest the importance of SRF-specific forms of TR. We also see strong advantages against outcome-based predictions and AIPW estimators, suggesting that the TR loss and the shared learning architecture is a boon to performance. These results are compatible with observations from previous work in the TR literature (Nie et al., 2021; Shi et al., 2019).

**Experiment 2: Does TR for count data improve estimation when count-valued outcomes are observed?** We evaluate whether TRESNET with the Poisson-specific TR, explained in Section 3, performs better than the mean-squared error (MSE) loss variant when the true data follows a Poisson distribution. This evaluation is important since our application consists of count data and the Poisson model is widely used to investigate the effects of PM$_{2.5}$ on health (Wu et al., 2020; Josey et al., 2022). We construct similar semi-synthetic datasets as in Experiment 1, but in this experiment the outcome-generating mechanisms samples from a Poisson distribution rather than a Gaussian distribution. The results of this experiment are clear–using the correct exponential family for the outcome model is crucial, regardless of whether TR is implemented. For these experiments, we used the spline-based variant of TRESNET.

## 6 APPLICATION: THE EFFECTS OF STRICTER AIR QUALITY STANDARDS

We implemented TRESNET for count data to estimate the health benefits caused by shifts to the distribution of PM$_{2.5}$ that would result from lowering the NAAQS—the regulatory threshold for the annual-average concentration of PM$_{2.5}$ enforced by the EPA.

**Data** The dataset is comprised of Medicare data[2] from 2000–2016, involving 68 million individuals. The data includes measurements on participant race/ethnicity, sex, age, Medicaid eligibility, and date of death, which are subsequently aggregated to the annual ZIP-code level. The PM$_{2.5}$ exposure measurements are extracted from an ensemble prediction model (Di et al., 2019). The confounders include measurements on meteorological information, demographics, and the socioeconomic status of each ZIP-code. Calendar year and census region indicators are also included to account for temporal and spatial trends. To compile our dataset, we replicated the steps and variables outlined by Wu et al. (2020).

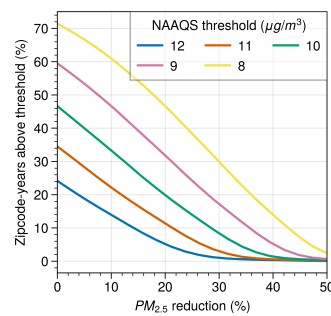

Figure 4: Fraction (%) of observed units remaining above PM$_{2.5}$ limit as a function of reduction (%) considering different NAAQS (current NAAQS is set at 12 μg/m³).

**Exposure shifts** We consider two types of PM$_{2.5}$ shifts, *cutoff shits* and *percent reduction shift*: each providing different perspectives and insights. The counterfactuals implied from these scenarios are illustrated in figures 2a and 2b, respectively. First, a *cutoff shift*, parameterized by a threshold $d$, encapsulates scenarios in which every ZIP-code year that exceeded some threshold are truncated to that maximum threshold. Mathematically, the shift is defined by the transformation $\tilde{A} = \min(A, c)$. To be more succinct, the exposure shift defines a counterfactual scenario. For this application, the threshold $c$ is evaluated at equally spaced points starting with 15 μg/m³ moving down to 6 μg/m³. We expect that at 15 μg/m³ there will be little to no reduction in deaths since $> 99\%$ of observations fall below that range. We can contemplate the proposed NAAQS levels through $d$ assuming that full compliance to the new regulation holds for incompliant ZIP-codes. The exposure shift should otherwise not affect already compliant ZIP-codes. Second, we onsider *percent reduction shifts*. This scenario assumes that all ZIP-code years reduce their pollution levels proportionally from their observed value. More precisely, the shift is defined as $\tilde{A} = A(1 - c)$. We considered a range of percent reduction shifts

---

[2]Access to Medicare data is restricted without authorization by the Centers for Medicare & Medicaid Services since it contains sensitive health information. The authors have completed the required IRB and ethical training to handle these datasets.

between $c \in (0, 50)\%$. We can interpret these shifts in terms of the NAAQS by mapping each percent reduction to a compliance percentile. For instance, Figure 4 shows that, under a 30% overall reduction in historical values, approximately 82% would comply with a NAAQS of 9 $\mu g/m^3$.

**Implementation** We implement TRESNET using varying-coefficient splines as in Section 5. We select a NN architecture using the out-of-sample prediction error from a 20/80% test-train split to choose the number of hidden layers (1-3 layers) and hidden dimensions (16, 64, 256). We found no evidence of overfitting in the selected models. To account for uncertainty in our estimations, we train the model on 100 bootstrap samples, each with random initializations, thereby obtaining an approximate posterior distribution. Deep learning ensembles have been previously shown to approximate estimation uncertainty well in deep learning tasks (Izmailov et al., 2021).

**Results** Figure 1 in the introduction presents the effects of shifting the $PM_{2.5}$ distribution at various cutoffs on the expected reduction to deaths. The slope is steeper at stricter/lower cutoffs, likely because lower cutoffs affect a larger fraction of the observed population and reduce the overall $PM_{2.5}$. For instance, figure Figure 1 shows that had no ZIP-code years exceeded 12 $\mu g/m^3$, the observed death counts would have decreased by around 1%. If the cutoff is lowered to 9 $\mu g/m^3$, then deaths could have fallen by around 4%. The slope becomes increasingly steeper as the $PM_{2.5}$ threshold is reduced, suggesting the increasing benefits of lowering the standard concentration level. Another way to interpret this result is to say that there is a greater gain to reducing mortality caused by $PM_{2.5}$ from lowering the concentration level from 10 to 8 $\mu g/m^3$ than there is from lowering it from 12 to 10 $\mu g/m^3$.

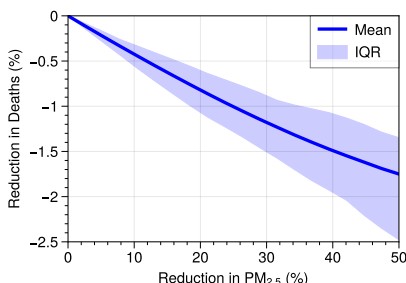

Figure 5: Estimated SRF of the total deaths (%) for different cutoffs.

The results of the percent-reduction shift are presented in Figure 5. The decrease in deaths is approximately linear with respect to the percent decrease in $PM_{2.5}$. As such, the SRF shows an approximate 0.5% decrease in deaths resulting from a 10% decrease in $PM_{2.5}$. This result is consistent with previous causal estimates of the marginal effect of $PM_{2.5}$ exposure on elder mortality (Wu et al., 2020). Percent reduction offers a complementary view to the cutoff shift response function.

# 7 DISCUSSION AND LIMITATIONS

We have made a significant stride in addressing the pressing public health question regarding the potential health benefits of lowering the NAAQS in the United States. In response to this question, we introduce the first causal inference method to utilize neural networks for estimating SRFs. Furthermore, we have extended this method to handle count data, which is crucial for our application in addition to other public health and epidemiology contexts. We acknowledge some limitations to our methodology. First, our uncertainty assessment of the SRF relies on the bootstrap and ensembling of multiple random seeds. While these methods are used often in practice, future research could explore the integration of TRESNET with Bayesian methods to enhance uncertainty quantification. Second, our application of the methodology focuses on exposure shifts representing complementary viewpoints to the possible effects of the proposed EPA rules on the NAAQS. However, it does not determine the most probable exposure shift resulting from the new rule's implementation, based on historical responses to changes in the NAAQS. Subsequent investigations should more carefully consider this aspect of the analysis. The assessment of annual average $PM_{2.5}$ levels at the ZIP-code level is based on predictions rather than on actual observable values, introducing potential attenuation bias stemming from measurement error. Nonetheless, previous studies on measurement error involving clustered air pollution exposures have demonstrated that such attenuation tends to pull the causal effect towards a null result (Josey et al., 2022; Wei et al., 2022). It is essential to recognize that the SRF framework places additional considerations on the analyst designing the exposure shift. This newfound responsibility can be seen as both a disadvantage and an advantage. However, it highlights the need for an explicit and meticulous statement of the assumptions underlying the considered exposure shifts in order to mitigate the potential misuse of SRF estimation techniques.

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

## A  BACKGROUND ON THE EIF

When we fit a statistical model to a dataset, each data point contributes to the estimated parameters of the model. The efficient influence function (EIF) effectively measures how sensitive an estimate is to the inclusion or exclusion of the individual data points. In other words, it tells us how much a single data point can influence the estimate of a causal effect. The term "efficient" refers to the property that, over the aggregate, the efficient influence function evaluated over the observed data provides the uniformly best statistically-efficient estimator of the associated causal effect under the nonparametric identifying assumptions that were made explicit in the main body of the manuscript (Kennedy, 2016). This property holds even when using machine learning methods (like neural networks) to estimate the nuisance parameters – in our case these nuisance parameters refer to the outcome regression model and the generalized propensity score model (Muñoz & Van Der Laan, 2012). An important distinction to make in this definition is that efficiency is an asymptotic property, meaning that it only holds as the sample size goes to infinity.

Another way to describe the efficient influence function is as the canonical gradient (Van der Laan et al., 2011) of the targeted parameter. Specifically, let $\{\mathbb{P}_t : t \in \mathbb{R}\}$ be a smooth parametric submodel such that $\mathbb{P}_0 = \mathbb{P}$ and denote its score function as $s_h(\boldsymbol{o}) = \frac{\mathrm{d}}{\mathrm{d}t}\big|_{t=h} \log \tilde{p}_t(\boldsymbol{o})$ where we

use $\tilde{p}_t(\boldsymbol{o}) = p_t(y|\boldsymbol{x}, a)\tilde{p}_t(a|\boldsymbol{x})p_t(\boldsymbol{x})$ and $p_t(\boldsymbol{o}) = p_t(y|\boldsymbol{x}, a)p_t(a|\boldsymbol{x})p_t(\boldsymbol{x})$ to denote the probability density function of $\mathbb{P}_t$. We also define $w_t(\boldsymbol{x}, a) = \tilde{p}_t(a|\boldsymbol{x})/p_t(a|\boldsymbol{x})$. The SRF estimand at each member of the submodel is defined as

$$\psi(\mathbb{P}_t) = \int \int \int y\tilde{p}_t(\boldsymbol{o}) \, \mathrm{d}\boldsymbol{o}.$$

For (9) to be the efficient influence function in a non-parametric model, one must show that it is *pathwise differentiable*, that is,

$$\frac{\mathrm{d}}{\mathrm{d}t}\Big|_{t=0} \psi(\mathbb{P}_t) = \int \varphi(\boldsymbol{o}; \psi, \mu, \boldsymbol{w})s_0(\boldsymbol{o})\tilde{p}(\boldsymbol{o}) \, \mathrm{d}\boldsymbol{o}. \tag{8}$$

This proof appears in the next section.

## B  TECHNICAL PROOFS

Before proving the main theorems in the paper, we show the following result regarding the identification of the $\psi$.

**Proposition 1.** *Suppose Assumption 2.1 holds. Then $\mu(\boldsymbol{X}, A) = \mathbb{E}[Y \mid \boldsymbol{X}, A]$. The right-hand side does not involve potential outcomes. As a corollary, $\psi = \mathbb{E}[\mu(\boldsymbol{X}, \tilde{A})]$ can be identified.*

*Proof.* Since the treatment and potential outcomes are independent conditional on $\boldsymbol{X}$ (unconfoundedness). By properties of the conditional expectation, we have

$$\begin{aligned}
\mu(\boldsymbol{x}, a) &= \mathbb{E}[Y^a \mid \boldsymbol{X} = \boldsymbol{x}] \\
&=^{(a)} \mathbb{E}[Y^a \mid \boldsymbol{X} = \boldsymbol{x}, A = a] \\
&= \mathbb{E}[Y^A \mid \boldsymbol{X} = \boldsymbol{x}, A = a] =^{(b)} \mathbb{E}[Y \mid \boldsymbol{X} = \boldsymbol{x}, A = a].
\end{aligned}$$

The identity (a) uses unconfoundedness (Assumption 2.1); (b) uses that $Y^A = Y$ (consistency). $\quad\square$

We now prove the three main theorems. The proof strategy follows the same pattern as Nie et al. (2021), with modifications to accommodate the SRF estimand and is mindful of the exponential family loss, which complicates the analytical form of the fluctuation term.

Suppose Assumptions 2.1 and 2.2 hold. Then the EIF of $\psi$ is given by

$$\varphi(\boldsymbol{O}; \psi, \mu, \boldsymbol{w}) = \boldsymbol{w}(\boldsymbol{X}, A)\left(Y - \mu(\boldsymbol{X}, A)\right) + \mu(\boldsymbol{X}, \tilde{A}) - \psi. \tag{9}$$

Furthermore, let $\hat{\mu} = \hat{\mu}(\mathbb{P}_n)$ and $\hat{\boldsymbol{w}} = \hat{\boldsymbol{w}}(\mathbb{P}_n)$ be estimators such that $\|\hat{\mu} - \mu\|_\infty = O_p(r_1(n))$ and $\|\hat{\boldsymbol{w}} - \boldsymbol{w}\|_\infty = O_p(r_2(n))$. Then $\|\mathbb{P}\varphi(\psi, \hat{\mu}, \hat{\boldsymbol{w}})\| = O_p(r_1(n)r_2(n))$. One of the conditions $\hat{\mu} = \mu$ or $\hat{\boldsymbol{w}} = \boldsymbol{w}$ suffice for $\mathbb{P}\varphi(\psi, \hat{\mu}, \hat{\boldsymbol{w}}) = \boldsymbol{0}$ to hold.

**Step 1: showing that $\varphi$ is the EIF**

*Proof.* Recall the notation from Appendix A. To show that $\varphi(\boldsymbol{o}; \psi, \mu, \boldsymbol{w})$ in Equation (9) is the EIF, we must show that

$$\frac{\mathrm{d}}{\mathrm{d}t}\Big|_{t=0} \psi(\mathbb{P}_t) = \int \varphi(\boldsymbol{o}; \psi, \mu, \boldsymbol{w})s_0(\boldsymbol{o})\tilde{p}(\boldsymbol{o}) \, \mathrm{d}\boldsymbol{o} \tag{10}$$

Due to the logarithmic transformation defining the score equation, we can factorize $s_t(\boldsymbol{o})$ into three parts with $s_t(\boldsymbol{o}) = s_{1,t}(y|a, \boldsymbol{x}) + s_{2,t}(a|\boldsymbol{x}) + s_{3,t}(\boldsymbol{x})$.

Starting on the left hand side of (10), under standard regularity conditions we have:

$$\begin{aligned}
\int y s_t(\boldsymbol{o})\tilde{p}_t(\boldsymbol{o}) \, \mathrm{d}\boldsymbol{o} &= \int y \frac{\mathrm{d}\tilde{p}_t(\boldsymbol{o})/\mathrm{d}t}{\tilde{p}_t(\boldsymbol{o})} p_t(\boldsymbol{o}) \, \mathrm{d}\boldsymbol{o} \\
&= \int y \frac{\mathrm{d}}{\mathrm{d}t}\tilde{p}_t(\boldsymbol{o}) \, \mathrm{d}\boldsymbol{o} \\
&= \frac{\mathrm{d}}{\mathrm{d}t} \int y\tilde{p}_t(\boldsymbol{o}) \, \mathrm{d}\boldsymbol{o},
\end{aligned} \tag{11}$$

which allows us to partition the pathwise derivative (i.e. the left-hand side of (10)) into:

$$
\begin{aligned}
\frac{\mathrm{d}}{\mathrm{d}t}\psi(\mathbb{P}_t) &= \int y s_t(\boldsymbol{o})\tilde{p}_t(\boldsymbol{o})\,\mathrm{d}\boldsymbol{o} \\
&= \int y(s_{1,t}(y|a,\boldsymbol{x}) + s_{2,t}(a|\boldsymbol{x}) + s_{3,t}(\boldsymbol{x}))\tilde{p}_t(\boldsymbol{o})\,\mathrm{d}\boldsymbol{o} \\
&= \int y s_{1,t}(y|a,\boldsymbol{x})\tilde{p}_t(\boldsymbol{o})\,\mathrm{d}\boldsymbol{o} + \int y s_{2,t}(a|\boldsymbol{x})\tilde{p}_t(\boldsymbol{o})\,\mathrm{d}\boldsymbol{o} + \int y s_{3,t}(\boldsymbol{x})\tilde{p}_t(\boldsymbol{o})\,\mathrm{d}\boldsymbol{o}.
\end{aligned}
\tag{12}
$$

For the next step of the proof, we will recursively use the two following identities. For any arbitrary function $g(\cdot)$, and for any two subsets of measurements $\boldsymbol{o}_1$ and $\boldsymbol{o}_2$ (e.g. $\boldsymbol{o}_1 = (a,\boldsymbol{x})$ and $\boldsymbol{o}_2 = y$), we have

$$
\int g(\boldsymbol{o}_1)\left\{\boldsymbol{o}_2 - \int \boldsymbol{o}_2 \tilde{p}_t(\boldsymbol{o}_2|\boldsymbol{o}_1)\,\mathrm{d}\boldsymbol{o}_2\right\}\tilde{p}_t(\boldsymbol{o})\,\mathrm{d}\boldsymbol{o} =^{(a)} 0
$$
$$
\int g(\boldsymbol{o}_1) s_t(\boldsymbol{o}_2|\boldsymbol{o}_1)\tilde{p}_t(\boldsymbol{o})\,\mathrm{d}\boldsymbol{o} =^{(b)} 0
\tag{13}
$$

Defining $\tilde{p}_0(\cdot) = \tilde{p}(\cdot)$ and $p_0(\cdot) = p(\cdot)$, we can then find the first term in (12) is equal to

$$
\begin{aligned}
\int y s_{1,t}(y|a,\boldsymbol{x})\,\mathrm{d}\tilde{p}_t(\boldsymbol{o})\,\mathrm{d}\boldsymbol{o}\Big|_{t=0} &= \int y s_{1,0}(y|a,\boldsymbol{x})\tilde{p}(\boldsymbol{o})\,\mathrm{d}\boldsymbol{o} \\
&=^{(b)} \int \{y - \mu(\boldsymbol{x},a)\}\, s_{1,0}(y|a,\boldsymbol{x})p(y|a,\boldsymbol{x})\tilde{p}(a|\boldsymbol{x})p(\boldsymbol{x})\,\mathrm{d}\boldsymbol{o} \\
&= \int w_t(\boldsymbol{x},a)\{y - \mu(\boldsymbol{x},a)\}\, s_{1,0}(y|a,\boldsymbol{x})p(y|a,\boldsymbol{x})p(a|\boldsymbol{x})p(\boldsymbol{x})\,\mathrm{d}\boldsymbol{o} \\
&=^{(a)} \int w_t(\boldsymbol{x},a)\{y - \mu(\boldsymbol{x},a)\}\{s_{1,0}(y|a,\boldsymbol{x}) + s_{2,0}(a|\boldsymbol{x})\}\,p(\boldsymbol{o})\,\mathrm{d}\boldsymbol{o} \\
&=^{(a)} \int w_t(\boldsymbol{x},a)\{y - \mu(\boldsymbol{x},a)\}\{s_{1,0}(y|a,\boldsymbol{x}) + s_{2,0}(a|\boldsymbol{x}) + s_{3,0}(\boldsymbol{x})\}\,p(\boldsymbol{o})\,\mathrm{d}\boldsymbol{o} \\
&= \int w_t(\boldsymbol{x},a)\{y - \mu(\boldsymbol{x},a)\}\, s_0(\boldsymbol{o})p(\boldsymbol{o})\,\mathrm{d}\boldsymbol{o}.
\end{aligned}
$$

For the second term, we have

$$
\begin{aligned}
\int y s_{2,t}(a|\boldsymbol{x})\,\mathrm{d}\tilde{p}_t(\boldsymbol{o})\,\mathrm{d}\boldsymbol{o}\Big|_{t=0} &= \int\int\int y s_{2,0}(a|\boldsymbol{x})p(y|a,\boldsymbol{x})\tilde{p}(a|\boldsymbol{x})p(\boldsymbol{x})\,\mathrm{d}y\,\mathrm{d}a\,\mathrm{d}\boldsymbol{x} \\
&= \int\int \mu(\boldsymbol{x},a) s_{2,0}(a|\boldsymbol{x})\tilde{p}(a|\boldsymbol{x})p(\boldsymbol{x})\,\mathrm{d}a\,\mathrm{d}\boldsymbol{x} \\
&=^{(b)} \int\int \left\{\mu(\boldsymbol{x},a) - \int \mu(\boldsymbol{x},a)\tilde{p}(a|\boldsymbol{x})\,\mathrm{d}a\right\} s_{2,0}(a|\boldsymbol{x})\tilde{p}(a|\boldsymbol{x})p(\boldsymbol{x})\,\mathrm{d}a\,\mathrm{d}\boldsymbol{x} \\
&=^{(b)} \int \left\{\mu(\boldsymbol{x},a) - \int \mu(\boldsymbol{x},a)\tilde{p}(a|\boldsymbol{x})\,\mathrm{d}a\right\}\{s_{1,0}(y|a,\boldsymbol{x}) + s_{2,0}(a|\boldsymbol{x})\}\tilde{p}(\boldsymbol{o})\,\mathrm{d}\boldsymbol{o} \\
&=^{(a)} \int \left\{\mu(\boldsymbol{x},a) - \int \mu(\boldsymbol{x},a)\tilde{p}(a|\boldsymbol{x})\,\mathrm{d}a\right\}\{s_{1,0}(y|a,\boldsymbol{x}) + s_{2,0}(a|\boldsymbol{x}) + s_{3,0}(\boldsymbol{x})\}\tilde{p}(\boldsymbol{o})\,\mathrm{d}\boldsymbol{o} \\
&= \int \left\{\mu(\boldsymbol{x},a) - \int \mu(\boldsymbol{x},a)\tilde{p}(a|\boldsymbol{x})\,\mathrm{d}a\right\} s_0(\boldsymbol{o})\tilde{p}(\boldsymbol{o})\,\mathrm{d}\boldsymbol{o}
\end{aligned}
$$

where we again center the integrand and apply the identities of (13). Finally, for the third term in (12) we have

$$
\begin{aligned}
\int y s_{3,t}(\mathbf{x}) \, \mathrm{d}\tilde{p}_t(\boldsymbol{o}) \, \mathrm{d}\boldsymbol{o}\Big|_{t=0} &= \int\int\int y s_{3,0}(\boldsymbol{x}) p(y|a,\boldsymbol{x}) \tilde{p}(a|\boldsymbol{x}) p(\boldsymbol{x}) \, \mathrm{d}y \, \mathrm{d}a \, \mathrm{d}x \\
&= \int \left\{ \int \mu(\boldsymbol{x},a)\tilde{p}(a|\boldsymbol{x}) \, \mathrm{d}a \right\} s_{3,0}(\boldsymbol{x}) p(\boldsymbol{x}) \, \mathrm{d}x \\
&=^{(b)} \int \left\{ \int \mu(\boldsymbol{x},a)\tilde{p}(a|\boldsymbol{x}) \, \mathrm{d}a - \psi \right\} s_{3,0}(\boldsymbol{x}) p(\boldsymbol{x}) \, \mathrm{d}x \\
&=^{(b)} \int \left\{ \int \mu(\boldsymbol{x},a)\tilde{p}(a|\boldsymbol{x}) \, \mathrm{d}a - \psi \right\} \left\{ s_{1,0}(y|a,\boldsymbol{x}) + s_{2,0}(a|\boldsymbol{x}) + s_{3,0}(\boldsymbol{x}) \right\} \tilde{p}(\boldsymbol{o}) \, \mathrm{d}\boldsymbol{o} \\
&= \int \left\{ \int \mu(\boldsymbol{x},a)\tilde{p}(a|\boldsymbol{x}) \, \mathrm{d}a - \psi \right\} s_0(\boldsymbol{o}) \tilde{p}(\boldsymbol{o}) \, \mathrm{d}\boldsymbol{o}.
\end{aligned}
$$

Combining these three terms above proves the condition in (10) holds, thus completing the proof. $\square$

**Step 2: showing the estimating equation and double robustness**

*Proof.* We can break down $\mathbb{P}\varphi$ in two terms,
$$
\mathbb{P}\varphi(\boldsymbol{\psi},\hat{\mu},\hat{\boldsymbol{w}}) = \mathbb{E}[\hat{\boldsymbol{w}}(\boldsymbol{X},A)(Y - \hat{\mu}(\boldsymbol{X},A))] + \mathbb{E}[\hat{\mu}(X,\tilde{A}) - \boldsymbol{\psi}]
$$
We can rewrite the first term as
$$
\mathbb{E}[\hat{\boldsymbol{w}}(\boldsymbol{X},A)(Y - \hat{\mu}(\boldsymbol{X},A))] = \mathbb{E}[\hat{\boldsymbol{w}}(\boldsymbol{X},A)(\mu(\boldsymbol{X},A) - \hat{\mu}(\boldsymbol{X},A))]
$$
For the second term, we can write
$$
\begin{aligned}
\mathbb{E}[\hat{\mu}(\boldsymbol{X},\tilde{A}) - \boldsymbol{\psi}] &= \mathbb{E}[\hat{\mu} - \mu(\boldsymbol{X},\tilde{A})] \\
&= \mathbb{E}[\boldsymbol{w}(\boldsymbol{X},A)(\hat{\mu}(\boldsymbol{X},\tilde{A}) - \mu(\boldsymbol{X},\tilde{A}))]
\end{aligned} \tag{14}
$$
Combining them, we have that
$$
\mathbb{P}\varphi(\boldsymbol{\psi},\hat{\mu},\hat{\boldsymbol{w}}) = \mathbb{E}[(\hat{\boldsymbol{w}}(\boldsymbol{X},A) - \boldsymbol{w}(\boldsymbol{X},A))(\mu(\boldsymbol{X},A) - \hat{\mu}(\boldsymbol{X},A))].
$$
From this expression, the conclusion of the theorem is evident. $\mathbb{P}\varphi(\boldsymbol{\psi},\hat{\mu},\hat{\boldsymbol{w}}) = \mathbf{0}$ if either $\hat{\mu} = \mu$ or $\hat{\boldsymbol{w}} = \boldsymbol{w}$. Further,
$$
\|\mathbb{E}[(\hat{\boldsymbol{w}}(\boldsymbol{X},A) - \boldsymbol{w}(\boldsymbol{X},A))(\mu(\boldsymbol{X},A) - \hat{\mu}(\boldsymbol{X},A))]\| \le \|\hat{\boldsymbol{w}} - \boldsymbol{w}\|_\infty \|\hat{\mu} - \mu\|_\infty = O_p(r_1(n)r_2(n)),
$$
completing the proof. $\square$

We can now prove the first theorem. In what follows we will adopt the notation $\mathbb{P}_n$ to denote the empirical distribution. For any function $f(\boldsymbol{O})$, we have that $\mathbb{P}_n f = \frac{1}{n} f(O_i)$ and $\mathbb{P}f = \mathbb{E}[f(\boldsymbol{O})]$.

**Theorem 1.** *Let $\epsilon$ denote a perturbation parameter and define*
$$
\begin{aligned}
\mathcal{L}^{tr}(\mu^{NN},\boldsymbol{w}^{NN},\epsilon)(\boldsymbol{O}) &= \Lambda(g(\mu^{NN}(\boldsymbol{X},A)) + \epsilon) - (g(\mu^{NN}(\boldsymbol{X},A)) + \epsilon)Y. \\
\mathcal{R}^{tr}(\mu^{NN},\boldsymbol{w}^{NN},\epsilon) &= \frac{1}{n}\sum_{i=1}^n \mathcal{L}^{tr}(\mu^{NN},\boldsymbol{w}^{NN},\epsilon)(\boldsymbol{O}_i).
\end{aligned} \tag{4}
$$
*Then $(\frac{\partial \mathcal{R}^{tr}}{\partial \epsilon})(\mu^{NN},\boldsymbol{w}^{NN},\epsilon) = 0$ iff $\frac{1}{n}\sum_{i=1}^n \boldsymbol{w}^{NN}(\boldsymbol{X}_i,A_i)(Y_i - g^{-1}(g(\mu^{NN}(\boldsymbol{X}_i,A_i)) + \epsilon))) = 0$.*

*Proof.* A key property of the exponential family of distributions and the associated link function $g$ that we will use are $\Lambda'(\eta) = \frac{d}{d\eta}\Lambda(\eta) = g^{-1}(E[Y|\eta])$ (McCullagh, 2019). Also note that $\frac{d}{d\epsilon}g(\tilde{\mu}^{NN}(\boldsymbol{X},A)) = \boldsymbol{w}^{NN}(\boldsymbol{X},A)$ for all $\mu^{NN}, \boldsymbol{w}^{NN}\epsilon$. These two observations and the chain rule give us

$$
\begin{aligned}
\mathbf{0} &= \frac{d}{d\epsilon}\mathcal{R}^{tr}(\hat{\mu},\hat{\boldsymbol{w}},\hat{\epsilon}) \\
&= \frac{1}{n}\sum_{i=1}^n \frac{\mathrm{d}}{\mathrm{d}\epsilon}\Big|_{\epsilon=\hat{\epsilon}} \left\{ \Lambda(g(\tilde{\mu}(\boldsymbol{X},A))) - Y g(\tilde{\mu}(\boldsymbol{X},A)) \right\} \\
&= \frac{1}{n}\sum_{i=1}^n \left\{ g^{-1}(g(\tilde{\mu}(\boldsymbol{X},A)))\hat{\boldsymbol{w}}(\boldsymbol{X},A) - Y\hat{\boldsymbol{w}}(\boldsymbol{X},A) \right\} = \frac{1}{n}\sum_{i=1}^n \hat{\boldsymbol{w}}(\boldsymbol{X},A)(\tilde{\mu}(\boldsymbol{X},A) - Y).
\end{aligned} \tag{15}
$$

The fact that $\hat{\psi}^{\text{tr}} = \frac{1}{n}\sum_{i=1}^{n}\tilde{\mu}(\boldsymbol{X}_i, \tilde{A}_i)$ satisfies the empirical estimating equation follows trivially from the fact that $\mathbb{P}_n\varphi(\hat{\psi}^{\text{tr}}, \tilde{\mu}, \hat{\boldsymbol{w}}) = \frac{1}{n}\sum_{i=1}^{n}\hat{\boldsymbol{w}}(\boldsymbol{X}, A)(Y - \tilde{\mu}(\boldsymbol{X}, A)) + \frac{1}{n}\sum_{i=1}^{n}\tilde{\mu}(\boldsymbol{X}_i, \tilde{A}_i) - \hat{\psi}^{\text{tr}}$. The first term is zero because of the above results while the last two terms cancel each other by definition. $\qquad\square$

**Theorem 2.** *Let $\mathcal{M}$ and $\mathcal{W}$ be classes of functions such that $\hat{\mu}, \mu \in \mathcal{M}$ and $\hat{\boldsymbol{w}}, \boldsymbol{w} \in \mathcal{W}$. Suppose assumptions 2.1 and 2.2 hold, and that the following regularity conditions hold: (i) $\|\mathcal{M}\|_\infty < \infty$, $\|\mathcal{W}\|_\infty < \infty$, $\|1/\mathcal{W}\|_\infty < \infty$; (ii) either $\hat{\mu} = \mu$, $\hat{\boldsymbol{w}} = \boldsymbol{w}$, or $\text{Rad}_n(\mathcal{M}) = O(n^{-1/2})$ and $\text{Rad}_n(\mathcal{W}) = O(n^{-1/2})$; (iii) the loss function in Equation (4) is Lipschitz; (iv) $\Lambda$ and $g$ are twice continuously differentiable. Then, the following statements are true:*

1. *The outcome and density ratio estimators of TR are consistent. That is, $\hat{\mu} \xrightarrow{p} \mu$ and $\hat{\boldsymbol{w}} \xrightarrow{p} \boldsymbol{w}$.*
2. *The estimator $\hat{\psi}^{\text{tr}}$ satisfies $\|\hat{\psi}^{\text{tr}} - \psi\|_\infty = O_p(n^{-1/2} + r_1(n)r_2(n))$ whenever $\|\hat{\mu} - \mu\|_\infty = O_p(r_1(n))$ and $\|\hat{\boldsymbol{w}} - \boldsymbol{w}\|_\infty = O_p(r_2(n))$.*

**Step 1: showing consistency of the outcome and density ratio models** This proof closely adapts Nie et al. (2021) for the ERF case. We will use the notation $\mathcal{R}_{\mathbb{P}}(\mu^{\text{NN}}, \boldsymbol{w}^{\text{NN}}) = \mathbb{P}\mathcal{L}(\mu^{\text{NN}}, \boldsymbol{w}^{\text{NN}})$ for the population risk, where $\mathcal{L}^{\text{NN}}$ is the loss function. Denote $\mu^*, \boldsymbol{w}^*$ as the population risk minimizers. We assume $\mu^* \in \mathcal{M}, \boldsymbol{w}^* \in \mathcal{W}$. The proof strategy is fairly standard and requires that the loss function be Lipschitz to ensure that the Rademacher complexity and boundedness assumptions extend the loss terms. The Lipschitz condition can often be relaxed with direct assumptions on the terms $\mathcal{L}(\mu^{\text{NN}}, \boldsymbol{w}^{\text{NN}})$ to have vanishing Rademacher complexity and boundedness (Wainwright, 2019).

*Proof.* We first show that the risk of the regularized parameters is not too different from the minimum population risk. Specifically, we show that

$$\mathcal{R}_{\mathbb{P}}^{\text{NN}}(\hat{\mu}, \hat{\boldsymbol{w}}) - \mathcal{R}_{\mathbb{P}}^{\text{NN}}(\mu^*, \boldsymbol{w}^*) = o(1) + O_p(n^{-1/2}). \tag{16}$$

To prove this fact, we first note that

$$\begin{aligned}
0 &\leq \mathcal{R}_{\mathbb{P}}^{\text{NN}}(\hat{\mu}, \hat{\boldsymbol{w}}) - \mathcal{R}_{\mathbb{P}}^{\text{NN}}(\mu^*, \boldsymbol{w}^*) \\
&\leq \mathcal{R}(\hat{\mu}, \hat{\boldsymbol{w}}) - \mathcal{R}^{\text{NN}}(\mu^*, \boldsymbol{w}^*)) + (\mathbb{P} - \mathbb{P}_n)\mathcal{L}^{\text{NN}}(\hat{\mu}, \hat{\boldsymbol{w}}) + (\mathbb{P}_n - \mathbb{P})\mathcal{L}^{\text{NN}}(\mu^*, \boldsymbol{w}^*).
\end{aligned} \tag{17}$$

The second and third terms are empirical processes. We now use the regularity assumptions on the Rademacher complexity. The order of the Rademacher complexity is preserved under Lipschitz transforms. Hence, under the assumption that the loss is Lipschitz, it follows that the class $\{\mathcal{L}^{\text{NN}}(\mu^{\text{NN}}, \boldsymbol{w}^{\text{NN}}): \mu^{\text{NN}} \in \mathcal{M}, \boldsymbol{w}^{\text{NN}} \in \mathcal{W}\}$ has a Rademacher complexity of order $O(n^{-1/2})$. Uniform boundedness is also preserved under Lispschitz transformations. Together, the vanishing Rademacher complexity and uniform boundedness imply the uniform law of large numbers, which in turn implies the convergence of the empirical process (Wainwright, 2019). Hence the last two terms are $O_p(n^{-1/2})$. We now bound the first term.

$$\begin{aligned}
\mathcal{R}^{\text{NN}}&(\hat{\mu}, \hat{\boldsymbol{w}}) - \mathcal{R}^{\text{NN}}(\mu^*, \boldsymbol{w}^*) \\
&= (\mathcal{R}^{\text{NN}} + \beta_n\mathcal{R}^{\text{tr}})(\hat{\mu}, \hat{\boldsymbol{w}}, \hat{\boldsymbol{\epsilon}}) - (\mathcal{R}^{\text{NN}} + \beta_n\mathcal{R}^{\text{tr}})(\mu^*, \boldsymbol{w}^*, \boldsymbol{0}) + \beta_n(\mathcal{R}^{\text{tr}}(\mu^*, \boldsymbol{w}^*, \boldsymbol{0}) - \mathcal{R}^{\text{tr}}(\hat{\mu}, \hat{\boldsymbol{w}}, \hat{\boldsymbol{\epsilon}})) \\
&\leq^{(a)} \beta_n(\mathcal{R}^{\text{tr}}(\mu^*, \boldsymbol{w}^*, \boldsymbol{0}) - \mathcal{R}^{\text{tr}}(\hat{\mu}, \hat{\boldsymbol{w}}, \hat{\boldsymbol{\epsilon}})) \\
&=^{(b)} \beta_n(\mathbb{P}_n\mathcal{L}_\mu(\mu^*) + O(1)). \\
&= \beta_n((\mathbb{P}_n - \mathbb{P})\mathcal{L}_\mu(\mu^*) + \mathbb{P}\mathcal{L}_\mu(\mu^*) + O(1)) \\
&=^{(c)} \beta_n(O_p(n^{-1/2})) + O(1) = o_p(1)
\end{aligned}$$

$$\tag{18}$$

Inequality (a) is due to $(\hat{\mu}, \hat{\boldsymbol{w}}, \hat{\boldsymbol{\epsilon}})$ being a minimizer for the regularized risk. Inequality (b) is the result of $\mathcal{R}^{\text{tr}}(\hat{\mu}, \hat{\boldsymbol{w}}, \hat{\boldsymbol{\epsilon}})$ being bounded since the exponential family of distributions is log-concave and $\mathcal{R}^{\text{tr}}(\mu^*, \boldsymbol{w}^*, \boldsymbol{0}) = \mathbb{P}_n\mathcal{L}_\mu(g(\mu^*))$. (c) uses the uniform law of large numbers from the Rademacher complexity and the uniform boundedness, and the fact that $\mathcal{L}$ is Lipschitz. Combining Equation (17) and Equation (18), we get $\mathcal{R}_{\mathbb{P}}^{\text{NN}}(\hat{\mu}, \hat{\boldsymbol{w}}) - \mathcal{R}_{\mathbb{P}}^{\text{NN}}(\mu^*, \boldsymbol{w}^*) = o_p(1)$.

The result now follows from observing that the population risk has a unique minimizer up to the reparameterization of the network weights. Hence, by regularity conditions, $\|\hat{\mu} - \mu\| = o_p(1)$ and $\|\hat{\boldsymbol{w}} - \boldsymbol{w}\| = o_p(1)$. $\qquad\square$

**Step 2: Proving convergence and efficiency of $\hat{\psi}^{\text{tr}}$**

*Proof.* Direct computation gives

$$
\begin{aligned}
\|\hat{\psi}^{\text{tr}} - \boldsymbol{\psi}\| &= \|\tfrac{1}{n} \textstyle\sum_{i=1}^n \tilde{\mu}(\boldsymbol{X}_i, \tilde{A}_i) - \boldsymbol{\psi}\| \\
&=^{(a)} \|\tfrac{1}{n} \textstyle\sum_{i=1}^n \{\tilde{\mu}(\boldsymbol{X}_i, \tilde{A}_i) + \hat{\boldsymbol{w}}(\boldsymbol{X}_i, A_i)(Y_i - \tilde{\mu}(\boldsymbol{X}_i, A_i))\} - \boldsymbol{\psi}\| \\
&=^{(b)} \|\mathbb{E}[\hat{\boldsymbol{w}}(\boldsymbol{X}, A)(Y - \tilde{\mu}(\boldsymbol{X}, A)) + \tilde{\mu}(\boldsymbol{X}, \tilde{A})] - \boldsymbol{\psi}\| + O_p(n^{-1/2}), \\
&=^{(c)} \|\mathbb{E}[\hat{\boldsymbol{w}}(\boldsymbol{X}, A)(\mu(\boldsymbol{X}, A) - \tilde{\mu}(\boldsymbol{X}, A)) + \tilde{\mu}(\boldsymbol{X}, \tilde{A})] - \boldsymbol{\psi}\| + O_p(n^{-1/2}), \\
&=^{(d)} \|\mathbb{E}[(\hat{\boldsymbol{w}}(\boldsymbol{X}, A) - \boldsymbol{w}(\boldsymbol{X}, A))(\mu(\boldsymbol{X}, A) - \tilde{\mu}(\boldsymbol{X}, A))]\| + O_p(n^{-1/2}),
\end{aligned}
\tag{19}
$$

where (a) is by the property of the targeted regularization, namely, $\frac{1}{n}\sum_{i=1}^n \hat{\boldsymbol{w}}(\boldsymbol{X}_i, A_i)(Y_i - \tilde{\mu}(\boldsymbol{X}_i, A_i)) = 0$; (b) is because of the uniform concentration of the empirical process, again using the vanishing Rademacher complexity and uniform boundedness; (c) integrates over $y$; (d) uses the definition of $\psi$ and the importance sampling formula with $\boldsymbol{w}$. Since the link function $g$ is continuously differentiable, invertible and strictly monotone, then by the mean value theorem there exists $\boldsymbol{\epsilon}' \in (0, \hat{\boldsymbol{\epsilon}})$ such that

$$
\tilde{\mu}(\boldsymbol{X}, A) = g^{-1}(g(\hat{\mu}(\boldsymbol{X}, A)) + \hat{\boldsymbol{\epsilon}}) = \hat{\mu}(\boldsymbol{X}, A) + (g^{-1})'(g(\hat{\mu}(\boldsymbol{X}, A)) + \boldsymbol{\epsilon}')\hat{\boldsymbol{\epsilon}}.
$$

From the uniform boundedness and smoothness of the link function, we have that $\hat{c} = (g^{-1})'(g(\hat{\mu}(\boldsymbol{X}, A) + \boldsymbol{\epsilon}') < C$ for some constant $C > 0$. Then, using the above result in the last term of Equation (19), we obtain

$$
\begin{aligned}
&\|\mathbb{E}[(\hat{\boldsymbol{w}}(\boldsymbol{X}, A) - \boldsymbol{w}(\boldsymbol{X}, A))(\mu(\boldsymbol{X}, A) - \tilde{\mu}(\boldsymbol{X}, A))]\| \\
&\leq \mathbb{E}[(\hat{\boldsymbol{w}}(\boldsymbol{X}, A) - \boldsymbol{w}(\boldsymbol{X}, A))(\mu(\boldsymbol{X}, A) - \hat{\mu}(\boldsymbol{X}, A))] + C\|(\hat{\boldsymbol{w}}(\boldsymbol{X}, A) - \boldsymbol{w}(\boldsymbol{X}, A))\hat{\boldsymbol{\epsilon}}\| \\
&\leq O_p(r_1(n)r_2(n)) + O_p(r_2(n))\|\hat{\boldsymbol{\epsilon}}\|
\end{aligned}
\tag{20}
$$

To complete the proof, we will show that $\|\hat{\boldsymbol{\epsilon}}\| = O_p(r_1(n)) + O_p(n^{-1/2})$. Letting $\hat{c}_i$ be as in the Taylor expansion above, we can re-arrange the targeted regularization condition such that

$$
\boldsymbol{0} = \frac{d}{d\boldsymbol{\epsilon}}\mathcal{R}^{\text{tr}}(\hat{\mu}, \hat{\boldsymbol{w}}, \hat{\boldsymbol{\epsilon}}) = \frac{1}{n}\sum_{i=1}^n \hat{\boldsymbol{w}}(\boldsymbol{X}, A)(\hat{\mu}(\boldsymbol{X}, A) - Y) + \frac{1}{n}\sum_{i=1}^n \hat{\boldsymbol{w}}(\boldsymbol{X}_i, A_i)\hat{c}_i\hat{\boldsymbol{\epsilon}}.
$$

Hence, we can write $\hat{\boldsymbol{\epsilon}}$ with the closed-form expression

$$
\hat{\boldsymbol{\epsilon}} = \arg\min_{\boldsymbol{\epsilon}} \mathcal{R}^{\text{tr}}(\hat{\mu}, \hat{\boldsymbol{w}}, \boldsymbol{\epsilon}) = \frac{n^{-1}\sum_{i=1}^n \hat{\boldsymbol{w}}(\boldsymbol{X}_i, A_i)(Y_i - \hat{\mu}(\boldsymbol{X}_i, A_i))}{n^{-1}\sum_{i=1}^n \hat{c}_i\hat{\boldsymbol{w}}(\boldsymbol{X}_i, A_i)^2}.
$$

Since the denominator is uniformly bounded in a neighborhood of the solution as in $\|1/\mathcal{W}\|_\infty < \infty$, and $g$ is strictly monotone and continuously differentiable, implies that $\hat{c}_i$ is uniformly lower bounded. Hence, there is $C' > 0$ such that

$$
\begin{aligned}
\|\hat{\boldsymbol{\epsilon}}\| &\leq C'\|n^{-1}\sum_{i=1}^n \hat{\boldsymbol{w}}(\boldsymbol{X}_i, A_i)(Y_i - \hat{\mu}(\boldsymbol{X}_i, A_i))\| \\
&\leq^{(a)} C'\|\mathbb{E}[\hat{\boldsymbol{w}}(\boldsymbol{X}, A)(Y - \hat{\mu}(\boldsymbol{X}, A))\| + O_p(n^{-1/2}) \\
&\leq^{(b)} O_p(r_1(n)) + O_p(n^{-1/2}),
\end{aligned}
\tag{21}
$$

where (a) uses the uniform concentration of the empirical process and (b) again uses the uniform boundedness of $\hat{\boldsymbol{w}}$. The proof now follows from combining equations (19), (20), and (21). $\square$

## C   ANOTHER EXAMPLE OF SRF VS ERF

The following examples show a simple case where the SRF and ERF estimands are different, thereby demonstrating why the ERF is not a useful estimate of the effect of an exposure shift. Consider a setting in which $\mu(\boldsymbol{X}, A) = AX$ with $X \sim N(0, 1)$, $A \sim N(X, 1)$. Now consider the exposure shift induced by $\tilde{A} = cA$ for some $c \in \mathbb{R}$. Using the SRF formulation, we find that $\psi = \mathbb{E}[\mu(\boldsymbol{X}, \tilde{A})] = \mathbb{E}[\mathbb{E}[\boldsymbol{X}(cA)|\boldsymbol{X}]] = c\mathbb{E}[\boldsymbol{X}^2] = c$. On the other hand, the ERF is $\xi(a) = \mathbb{E}[\mu(\boldsymbol{X}, A)|A = a] = a\mathbb{E}[\boldsymbol{X}] = 0$ for all $a \in \mathbb{R}$. Therefore, estimators of the two estimands return two different estimates. Moreover, the ERF is identically zero for every treatment value. Thus, it cannot be used to approximate the value of the effect of the exposure shift, even when $\psi$ is correctly specified.

## D    HARDWARE/SOFTWARE/DATA ACCESS

We ran all of our experiments both in the simulation study and the application section using Pytorch
(**?**) on a high-performance computing cluster equipped with Intel 8268 "Cascade Lake" processors.
Due to the relatively small size of the datasets, hardware limitations, and the large number of sim-
ulations required, we did not require the use of GPUs. Instead we found that using only CPUs run
in parallel sufficed. Reproducing the full set of experiments takes approximately 12 hours with 100
parallel processes, each with 4 CPU cores and 8GB of RAM. Each process runs a different random
seed for the experiment configuration.

The code for reproducibility is provided on the submission repository along with the data sources
for the simulation experiments. The datasets for these experiments were obtained from the public
domain and were adapted from the GitHub repositories shared by Nie et al. (2021) and Bica et al.
(2020) as explained in the experiment details section. The data for the application was purchased
from `https://resdac.org/`. Due to a data usage agreement and privacy concerns, manipula-
tion of these data requires IRB approval under which the authors have completed the training and
for which reason the data cannot be shared with the public.

