# OpenReview forum: "Causal Estimation of Exposure Shifts with Neural Networks: Evaluating the Health Benefits of Stricter Air Quality Standards in the US"
_ICLR.cc/2024/Conference — Submitted to ICLR 2024_

### Official Review · Reviewer_Ka1h · 2023-10-30

**Soundness:** 3 good
**Presentation:** 3 good
**Contribution:** 3 good
**Rating:** 6
**Confidence:** 4

**Summary:**

The authors consider so-called shift-response function (SRF) estimation with neural network methods, which is motivated by a policy-relevant question in public health, and statistical robustness and efficiency consideration. They apply their method to data consisting of 68 million individuals and 27 million deaths across the U.S. to estimate the causal effect from revising the US National Ambient Air Quality Standards (NAAQS) for PM2.5 from 12 μg/m3 to 9 μg/m3.

**Strengths:**

The problem is well motivated and the application is interesting.

**Weaknesses:**

I am not fully convinced about this causal estimand. In Section C's example, why $c$ is a better number than 0? Can the authors clarify?

Can the shift of treatment be stochastic?

Some key references on doubly robust estimator and efficiency of causal effect estimation are missing, for example, Robins's work.

Should $\mu(x,a)$ is (1) $\mu(X,\bar A)$?

There are typos, for example, in Section C, EIF should be ERF; some references are broken.

**Questions:**

Please refer to Weakness.

---

> ### Author Response · Authors · 2023-11-20
>
> Thank you for your accurate summary and comments regarding the well-motivated application. We address your questions below.
>
> 1. The application generally sets the sensible choice of shift value $c$. Using the example of Section C asked by the reviewer, $c=1.1$ asks the question of what would the average outcome be under a uniform 10% increase in the treatment; $c=0$ is a degenerate case in which everybody receives treatment $\tilde{A}=0$; $c=1$ corresponds simply to the observed average outcome without counterfactuals. It is generally the role of the scientist or policymaker to choose the shift value under consideration. For example, in the application solved in our paper, it is sensible to ask what the effect reducing or increasing by PM 2.5 air pollution by 10% would be (Fig. 2b). The cutoff shift, also used in the application, is another example (Fig. 2a) in which a maximum threshold is allowed. The relevant shifts, $8,9,10,11 \mu g/m^3$, analogous to $c$ with a different type of shift, are set by the US Environmental Protection Agency.
> 2. There is no problem with the shift being stochastic as our paper is currently formulated. Our key assumption is that we observe pairs $(A, \tilde{A})$, with the latter measurement potentially being a stochastic draw. This flexible setup allows us to address the application problem. We point out to the reviewer the clarity improvements that we made to Section 2 of the revision introducing the estimand and clarifying the reviewer's question.
> 3. The revision has added various relevant works by Robin. However, do note that these previous works concern traditional treatment effect estimation and do not cover the case of estimating the effect of an exposure shift (SRF). Our paper addresses this problem building upon the targeted regularization framework (Nie et al., ICLR 2021; Shi et al., NeurIPS 2019). The SRF scenario is sufficiently different from traditional causal effect estimation, requiring additional theory and methods.
> 4. It should be $\mu(X, \tilde{A})$.
> 5. We have fixed this typo and the broken reference in the revision. Thank you for your careful review.

---

> > ### Author Response · Authors · 2023-11-21
> >
> > Dear reviewer, thanks again for your review. As the author/discussion period ends tomorrow, we would be extremely grateful if you could let us know if your questions and concerns have been resolved. Particularly your question (1) about the estimand.

---

> > > ### Comment · Reviewer_Ka1h · 2023-11-22
> > >
> > > Thank you for your clarification. Regarding 2, can I tell if the exposure shift is deterministic or stochastic from the observed $(A,\tilde A)$? And is it relevant to the scientific question of interest?

---

> > > > ### Author Response · Authors · 2023-11-22
> > > >
> > > > In the scientific problem addressed in the paper, the exposure shift is deterministic *conditional* on $A$ since we consider the effects of two types of shifts: the percent shift reduction $\tilde{A}=(1 - c)A$ where $c$ is the reduction factor; and a cutoff shift $\tilde{A}=min(A, c)$ where $c$ is the maximum allowed value. The significance of these shifts is explained in Section 6. See also Fig. 2a and 2b. It must be remarked that even though the transform is deterministic conditional on $A$, the framework would still correspond to what has been called  *stochastic intervention* in the causal inference literature (c.f. Diaz & Van der Laan 2012) since in traditional causal effect estimation $\tilde{A}$ is assumed fixed at the same value for all units in the exposure-response function (see Fig 2c), while in our setup it is different in each unit. Thus, marginally, when not conditioning on $A$, it is stochastic.

---

### Official Review · Reviewer_KwaZ · 2023-11-01

**Soundness:** 3 good
**Presentation:** 4 excellent
**Contribution:** 3 good
**Rating:** 6
**Confidence:** 3

**Summary:**

This paper considers the problem of shift response function estimation, with applications
to evaluating the health benefits of stricter air quality standards in the US.

The proposed method falls into the framework of AIPW, where both the outcome model and the
propensities are trained via neural networks with regularization terms. The authors provide
theoretical results supporting that the resulting estimator indeed is double robust and efficient. The method
is evaluated on synthetic data and applied to the evaluation of the health benefit of stricter
air quality standards.

**Strengths:**

1. The paper is very well-written: it is concise and contains sufficient details.
2. I think this work is a good combination of application and theory. Motivated by an important
practical question, the authors formulate it as a mathematical problem, providing solutions
backed with theoretical results.

**Weaknesses:**

The theoretical result is not particularly surprising given the existing literature on double-robustness and efficiency (although I do like the application side of this work).

**Questions:**

1. I am in general curious about the reason for choosing neural networks to fit the propensities and
the outcome model. How do they compare with, say, tree-based methods?
2. In many scenarios, the multiple shifts are being considered simultaneously, should there be adjustment
for the multiplicity?
3. Some minor points:

  (a) in equation (1), should $\mu(x,a)$ be $\mu(X,\tilde{A})$?

  (b) there is a missing reference at the bottom of page 5.

---

> ### Author Response · Authors · 2023-11-20
>
> Thank you for your positive remarks highlighting the application’s strengths and the paper’s clarity.
>
> As a point of clarification, our framework builds upon previous literature on targeted regularization (TR), particularly Nie et al. (ICLR 2021). Previous results in the TR literature and our theoretical results may not be entirely surprising from the point of view of the long tradition of double robust estimation in causal inference. Nonetheless, such theoretical results are required to ensure correctness in the novel framework of TR. Notice as well that compared to previous work in the TR literature, we address the problem of estimating the effect of an exposure shift (SRF). This scenario is sufficiently different that it requires additional theory and methods (e.g., see Diaz et al. (2021) and Diaz & Van der Laan (2012)). Our paper rigorously extends the TR literature to SRFs while solving an application of urgent importance, providing the first application of targeted regularization in a critical public health application.
>
> We address your questions below.
>
> 1. Earlier research on TR by Nie et al. (ICLR 2021) and Shi et al. (NeurIPS 2019) gave experimental results suggesting that TR can outperform tree-based methods using the AIPW, TMLE, BART and causal forest frameworks. Tree-based methods are incompatible with TR since they are inherent to gradient-based methods. However, we believe that using neural networks or tree-based methods is ultimately a matter of convenience and the nature of the data, as with many decisions in machine learning. Tree methods are generally easier to train and are more resilient to overfitting, but neural networks can perform greatly when correctly specified and are compatible with other forms of learning and regularization. Multi-modal systems with neural network building blocks are gaining importance (text, images, time series).
> 2. Eq. (8) includes a normalization by $|\tilde{P}|$, the number of exposure shifts considered. Is this the type of normalization considered by the reviewer?
> 3. Yes, it should be $\mu(X, \tilde{A})$. We have fixed the missing citations; thank you.
>
> We also point the reviewer to the various clarity improvements in the revision, which are marked with colored lettering.

---

> > ### Author Response · Authors · 2023-11-21
> >
> > Dear reviewer, thanks again for your review. As the author/discussion period ends tomorrow, we would be extremely grateful if you could let us know if your questions and concerns have been resolved. Particularly your question (2).

---

> > > ### Comment · Reviewer_KwaZ · 2023-11-22
> > >
> > > I would like to thank the authors for their response. I find my concerns addressed.

---

> > > > ### Author Response · Authors · 2023-11-22
> > > >
> > > > Thank you for your response.

---

### Official Review · Reviewer_vbhD · 2023-11-01

**Soundness:** 2 fair
**Presentation:** 2 fair
**Contribution:** 2 fair
**Rating:** 3
**Confidence:** 3

**Summary:**

The authors propose a neural network-based method, termed Targeted Regularization for Exposure Shifts with Neural Networks (TRESNET), to perform shift-response function (SRF) estimation for determining the causal effect of policy changes. The specific focus is on the effect of the proposed revision to the US National Ambient Air Quality Standards on mortality rates. The proposed TRESNET method introduces a targeted regularization loss tailored for SRF estimation, which ensures double robustness and asymptotic efficiency.

**Strengths:**

1. The paper addresses a meaningful real-world issue – evaluating the health benefits of air quality standards.

2. The proposed TRESNET method introduces a targeted regularization loss tailored for SRF estimation, which ensures double robustness and asymptotic efficiency.

**Weaknesses:**

1. The problem of this paper was not well presented. For example, the key concept of exposure shift is very confusing. The notation $\tilde{A}$ is used first without definition in Section 2. How is the potential outcome framework defined under $\tilde{A}$? The equation (1) is also problematic as it should be $a\sim \tilde{p}(\tilde{A}|X)$.

2. The assumptions of this work also need more justifications. It looks like all the causal identification assumptions are based on the original treatment $A$ except the positivity assumption. Since the efficient function also contains $\mu(X, $\tilde{A}$)$, would more assumptions on $\tilde{A}$ be needed like SUTVA?

3. The proposed method is not new compared with the semiparametric literature and causal inference, by considering double robustness and the density ratio of two propensities. Please find the references below and justify them.

- Yang, Shu, and Peng Ding. "Combining multiple observational data sources to estimate causal effects." Journal of the American Statistical Association (2019).
- Kallus, Nathan, and Xiaojie Mao. "On the role of surrogates in the efficient estimation of treatment effects with limited outcome data." arXiv preprint arXiv:2003.12408 (2020).

4. The theoretical connections between Section 3 and Section 4 are weak. There are many theoretical works related to using neural network methods for nuisance function estimation. The authors may consider the following reference to complete the gap.

- Farrell, Max H., Tengyuan Liang, and Sanjog Misra. "Deep neural networks for estimation and inference." Econometrica 89.1 (2021): 181-213.

5. Since the efficient influence function is derived, given Theorem 2, I am curious why not continue to get the asymptotic normality of the proposed effect? Specifically, the authors are using the asymptotic normal formula in their simulations. Or if the authors think it is challenging, why can you use this result directly in the simulation?

6. Since the double robustness is one major advantage of the proposed method, it is better to conduct simulation studies to reflect this property.

7. Mics: The reference at the bottom of page 5 is missing.

**Questions:**

See questions in *Weaknesses*.

---

> ### Author Response · Authors · 2023-11-20
>
> Thank you for highlighting that our application paper addresses a meaningful real-world issue using targeted regularization.
>
> Our paper was submitted to the conference in the *application* primary area and provides the first application of targeted regularization in an open public health problem requiring estimating the effects of exposure shifts.  We provide the necessary methodological extensions around the application. While our focus in the application, we aim to present rigorously such methodological extensions. The revision will include the necessary improvements in color lettering. We will address your itemized comments in the order they appear.
>
> 1. Section 2 in the original manuscript introduced $\tilde{A}$ as a variable to denote the shifted exposure, and examples were provided after eq (1). Nonetheless, the revision has improved and expanded Section 2, paragraph 3, to improve clarity. We hope the reviewer finds the updated version clearer.
> 2. Theorem 2 in the original manuscript imposed the additional restrictions required on the class of admissible density ratio functions $w(x,a)=\tilde{p}(a|x)/p(a|x)$. To improve clarity in the revision, we modified Assumption 2.2 and its explanation to give more intuitive and direct conditions tied to the regularity of the density ratio $w$ used for SRF estimation. Note that we provide mathematical proofs of all statements, justifying their definition. Assumption 2.2 is now in line with previous related work, specifically, Diaz & van der Laan (2012).
> 3. We reviewed the suggested citations, Shu & Ding (2019) and Kallus & Mao (2020). However, we did not find their immediate relevance to the problem addressed in our paper and the literature we are building upon. These works use auxiliary data (which we don’t use), and their aim is traditional treatment effect estimation comprising the average treatment effect (ATE) for binary treatments and the exposure-response function (ERF) for continuous treatments, as explained in Section 2 and Appendix C. Our paper addresses an application in which such traditional effect estimation is not informative, and estimating the effect of an exposure shift is necessitated instead. There are formulas involving density ratios in these works, but they have a different purpose and are defined in terms of other quantities, namely causal estimation with partial unmeasured confounding and causal inference with surrogate outcomes.
> 4. The revision includes the recommended citation by Farrel et al. (2021) in Section 4 for useful background, as suggested. Notice that the architectures discussed there concern traditional causal effect estimation (ATE and ERFs): they cannot be used to estimate the ratio between generalized propensity scores ($w$ in our paper), nor do they allow for double robustness of SRFs, as our proposed method does. The SRF scenario requires additional theory and methods beyond traditional causal effect estimation (Diaz et al. 2021; Diaz & Van der Laan, 2012). The extended paragraph also indicates that we based our architecture on previous work in targeted regularization (Nie et al. 2021), focusing on the required extensions for density ratio estimation.
> 5. The targeted regularization literature that our paper builds upon has yet to establish theoretical results and experiments concerning asymptotic normality. Notice that these works have been published at similar venues (Nie et al., ICLR 2021; Shi et al. NeurIPS 2019). We agree that future work in targeted regularization methods and theory should prioritize the investigation of these properties, but it is beyond the scope of our application paper.
> 6. Similarly, our evaluation criterion aligns with the literature we build upon. As in our answer to 5, we agree that double robustness could be examined with specific experiments, but that would go beyond the scope of our application paper and would require a paper on itself to examine the properties of *all* targeted regularization methods that have been proposed.
> 7. We have fixed the broken reference; thank you for identifying it.

---

> ### Author Response · Authors · 2023-11-21
>
> Dear reviewer, thanks again for your dedicated review. As the author/discussion period ends tomorrow, we would be extremely grateful if you could let us know if your questions and concerns have been resolved.  Your review raised concerns about the problem presentation. The revision addresses this point and your other technical remarks. We would greatly appreciate it if you let us know of other questions you may have. We are understanding of the incredibly short timeline.

---

### Official Review · Reviewer_xEjj · 2023-11-11

**Soundness:** 2 fair
**Presentation:** 2 fair
**Contribution:** 2 fair
**Rating:** 5
**Confidence:** 3

**Summary:**

This study provides a strong framework for treatment effect estimation based on the semiparametric theory. The authors develop a novel optimization problem for treatment effect estimation and show the asymptotic properties.

**Strengths:**

My major curiosity lies in the proof of Theorem 2. As discussed in the literature of double machine learning (cf. Chernozhukov et al. (2018)), to attain $\sqrt{n}$-convergence of semiparametric estimators, we usually impose the Donsker condition for (nonparametric) nuisance estimators. However, it seems that the authors do not impose such assumptions. I am checking the proof, but could the authors provide intuitive reasons for the results? If this result is true, I believe that this is the theoretical strength of this study.

The above is also my concern because the posited assumptions are too weak to show the results. We usually impose some properties such as smoothness on the nuisance estimators to discuss convergence rates. Even if we obtain desirable convergence rates, neural network models usually do not satisfy the Donsker conditions. Therefore, I am afraid of missing assumptions or errorrness in the proof (I need to confirm the proof but have not yet done it...).

**Weaknesses:**

See above.

**Questions:**

- Is Assumption 2.2 sufficient? If $p(a|x) \propto 1/n$ for some $a$, then the results do not hold, I think $p(a|x)$ should be lower bounded by a positive constant independent of $n$.
- In Theorem 2, what does $\to$ indicate? Convergence of non-random variables or convergence in probability?
- In Theorem 2, what does $O(r_1(n))$ in $\|\hat{\mu} - \mu\|_\infty$ mean? Should it be $O_P$?
- Sugiyama et al. (2012) discussed the density-ratio estimation, and its interest differs from this study. Furthermore, it has been known that Eq. (8) can be used to estimate the propensity score. I think the citation may not be appropriate.
- Does this study relate to automatic debiased learning proposed by Chernozhukov et al. (2022)?
- "for some function $\eta:\mathcal{X}\times\mathcal{A}\to\mathbb{R}$" should be for "for some measurable function $\eta:\mathcal{X}\times\mathcal{A}\to\mathbb{R}$"?
- Does the shift-response function is the same as the standard average treatment effect?
- Some citations are missing.

**Details Of Ethics Concerns:**

None.

---

> ### Author Response · Authors · 2023-11-20
>
> Thank you for the positive remarks in your summary, careful reading, and thoughtful questions.
>
> We included improvements and clarifications in the revised manuscript in color lettering. The reviewer's comments helped solve notation issues in the convergence statements, particularly regarding convergence in probability. We would like to also emphasize that our paper's primary submission area is application. Our key contribution is providing the first targeted regularization (TR) application to a critical open problem in public health requiring estimating the effects of an exposure shift (SRFs). We provide the necessary methodological extensions around the application.
>
> We address your questions below in order:
> 1. Additional regularity conditions stabilizing the density ratios were included as part of Theorem 2 (T2) since they weren’t required to present the key concepts. Specifically, assumption (ii) of T2 requires that the density ratio class $\mathcal{W}$ and its reciprocal have bounded $\infty$-norm. In the revision, we have reworded Assumption 2.2 to more clearly indicate the needed positivity assumptions on the density ratio function $w$. Assumption 2.2 is now in line with previous related work, specifically, Diaz & van der Laan (2012).
> 2. The arrow denotes convergence in probability (updated in the revision).
> 3. Yes (updated in the revision).
> 4. To clarify, we are not estimating a propensity score as in traditional treatment effect estimation (see also our response to Q7 below). The SRF framework requires estimating the conditional density ratio between two generalized propensity scores (Diaz & Van der Laan, 2012; Diaz et al., 2021). We originally cited the review paper by Sugiyama et al. (2012) on density ratio estimation since it was informative while developing our methods, however, it is indeed not required as we are citing other related work using the same density ratio formula.
> 5. The work by Chernozhukov et al. (2022) does not provide any guarantees specific for estimating the effect of an exposure shift (SRF) as our method does. More broadly, there are connections between the double machine learning literature and doubly robust estimation with the EIF, which is used by the TR framework that our paper builds upon. However, these connections remain under-explored. As far as we know, these connections have not been discussed in the TR literature.
> 6. We intentionally avoided measure-theoretical language since the targeted regularization literature we build upon has proved the necessary formal results without resorting to it. We understand the advantages of additional formality, but we also sought to align our notation with the literature.
> 7. SRFs are different from traditional treatment effects, comprising the average treatment effect (ATE) for binary treatment and the exposure-response function (ERF) for continuous treatments, as explained in Section 2 and Appendix C. SRFs allow for a different family of problems but require additional theory and methods (e.g., see Diaz et al. (2021) and Diaz & Van der Laan (2012)). We have improve the presentation of SRFs and examples in Section 2, paragraph 3, of the revision.
> 8. Thank you for your careful reading. We fixed typos and missing references in the revision.
>
> Regarding your question about Donsker classes, notice that we impose conditions on the Rademacher complexity and boundedness of the function classes in T2. These conditions suffice for a uniform law of large numbers, ensuring convergence. Note that our regularity assumptions in terms of Rademacher complexity are based on the targeted regularization literature we build upon, specifically, Nie et al. (ICLR 2021), who also do not use Donsker classes. Assumptions on Donsker classes are stronger; they ensure a limiting distribution and are required to prove asymptotic normality. As the reviewer pointed out, relying on weaker conditions strengthens the approach, considering our work is an application paper and asymptotic normality has yet to be established in the theory of targeted regularization that our paper builds upon.

---

> > ### Author Response · Authors · 2023-11-21
> >
> > Dear reviewer, thanks again for your thoughtful initial review. As the author/discussion period ends tomorrow, we would be very grateful if you could let us know if your questions have been mostly resolved or if you have any additional questions.  We understand that the timeline is extremely short and that there are multiple reviewing commitments.

---

### Meta-Review · Area_Chair_vf2m · 2023-12-12

**Metareview:**

The paper needs to improve upon the formal presentation of their main results and how they fit into the broader literature of semi-parametric inference.  The methodology seems incremental compared to prior work combining neural training and targeted regularization ideas.

**Justification For Why Not Higher Score:**

Many reviewers were negative

**Justification For Why Not Lower Score:**

N/A

---

### Decision · Program_Chairs · 2024-01-16

Reject